# Exploring and Exploiting the Asymmetric Valley of Deep Neural Networks

**Xin-Chun Li**[1,2], **Jin-Lin Tang**[1,2], **Bo Zhang**[1,2], **Lan Li**[1,2], **De-Chuan Zhan**[1,2]

[1] School of Artificial Intelligence, Nanjing University, Nanjing, China
[2] National Key Laboratory for Novel Software Technology, Nanjing University, Nanjing, China
{lixc, tangjl, zhangb, lil}@lamda.nju.edu.cn, zhandc@nju.edu.cn

## Abstract

Exploring the loss landscape offers insights into the inherent principles of deep neural networks (DNNs). Recent work suggests an additional asymmetry of the valley beyond the flat and sharp ones, yet without thoroughly examining its causes or implications. Our study methodically explores the factors affecting the symmetry of DNN valleys, encompassing (1) the dataset, network architecture, initialization, and hyperparameters that influence the convergence point; and (2) the magnitude and direction of the noise for 1D visualization. Our major observation shows that the *degree of sign consistency* between the noise and the convergence point is a critical indicator of valley symmetry. Theoretical insights from the aspects of ReLU activation and softmax function could explain the interesting phenomenon. Our discovery propels novel understanding and applications in the scenario of Model Fusion: (1) the efficacy of interpolating separate models significantly correlates with their sign consistency ratio, and (2) imposing sign alignment during federated learning emerges as an innovative approach for model parameter alignment.

## 1 Introduction

The massive number of parameters and complex structure of deep neural networks (DNNs) have catalyzed extensive research to mine their underlying mechanics [55, 34, 42, 62, 71]. Visualizing and exploring the loss surfaces of DNNs is the most intuitive way [43, 18], which has ignited many interesting findings, such as the monotonic linear interpolation [22, 19, 66] and linear mode connectivity [22, 14, 21, 15, 65, 2]. Loss landscape visualization has also been applied to show the optimization trajectory [53, 33, 35], understand the effectiveness of Batch Normalization [32, 61], BERT [11, 23], deep ensemble [17, 30, 21], and so on.

Perturbation analysis around the local minima of DNNs [7, 63], i.e., the shape of the valleys they reside in, is a very popular research topic. The concept of flat minima was originally proposed by [28], who defines the size of the connected region around the minima where the loss remains relatively unchanged as flatness. Subsequent studies debate whether the flat or sharp minima could reflect the generalization ability [38, 43, 36, 57, 16, 12, 41, 3]. The previous works constrain the valley shape to be symmetric, while recent work points out that not all DNN valleys are flat or sharp, and there also exist asymmetric valleys [25], which *has not been systematically studied as far as we know*.

This paper in-depth analyzes the factors that may affect the valley symmetry of DNNs. Previous work's analysis of valley shape primarily utilizes the 1D interpolation of $\theta_f + \lambda\epsilon$, where $\theta_f$ represents the minima solution and $\epsilon$ denotes a random noise. As shown in Fig. 1, we believe that the valley symmetry depends both on the convergence solution and noise, with each of them being influenced by some factors. *The most significant innovation in our research is considering the effect of noise direction on valley visualization, as previous work has simply taken the Gaussian noise.*

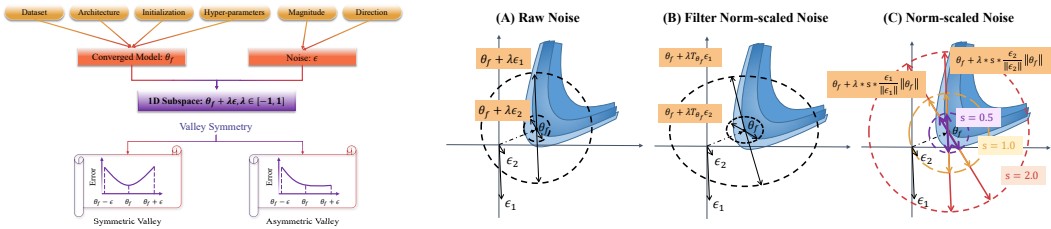

Figure 1: The illustration of investigated factors that could affect the valley symmetry. The $\epsilon$ matters a lot.

Figure 2: The illustration of different visualization methods for 1D visualization. The norm-scaled noise *unifies the magnitude* of various noise *without changing their directions*.

Specifically, we start by comprehensively and carefully determining the visualization method to plot the valley shape, which could influence the conclusion significantly [31, 43, 41]. We finally take the Norm-Scaled (NS) visualization method [31] that normalizes the noise $\epsilon$ and further scales it to $\|\theta_f\|$ for better determining the plot range of $\lambda$, where the direction of raw noise is not changed. Then, after investigating 7 common noise directions and 6 special ones, we conclude: *the degree of sign consistency between the noise and the convergence solution should be a determining factor for asymmetry*. This phenomenon is basically insensitive to the utilized datasets. Next, we focus on the impact of the network architecture with or without Batch Normalization (BN) [32], indicating that the BN initialization also impacts the valley symmetry. Finally, different hyperparameters lead to solutions with various valley widths but show no asymmetry consistently.

Aside from empirical observations, theoretical insights for our interesting findings are provided. We first declare that adding sign-consistent noise to parameters may have a larger probability of keeping activating the neurons or keeping the overwhelming score in classification tasks. Then, we show that the trace of the Hessian matrix along the sign-consistent direction is smaller, implying a flatter region. The above findings inspire applications in the fields of model fusion [47, 68, 33, 2]. This paper first explains why model aggregation based on pre-trained models often leads to performance improvements, i.e., the success of model soups [68], and then proposes constraining the sign of DNN parameters in federated learning [54, 73, 50] to facilitate aggregation.

Our novel contributions can be summarized as (1) *exploring the valley shape under different noise directions that have not been studied yet*; (2) *proposing that the flat region could be expanded along the direction that has a higher sign consistency with the convergence solution*; (3) *pointing out the influence of BN and its initialization on valley symmetry*; (4) *presenting theoretical insights to explain our interesting finding*; (5) *explaining and inspiring effective algorithms in model fusion*.

## 2  Related Works

**Exploring the Valley Shape of DNNs**. The valley around a minima solution has originally been viewed as flat or sharp [28], and the large batch size training may lead to sharp minima with poor generalization [38]. However, [12] declares that two solutions that are scale-invariant in performance may lie in regions with significantly different flatness. [43] verifies the findings of [12] by applying varying weight decay to small and large batch training, leading to results contrary to [38]. With the filter-normalized plots, [43] again observes that sharpness correlates well with generalization error. Later work also shows that the sharpness calculation should be dependent on the parameter scale [41] or the dataset [3]. Whether valley width reflects the generalization is still inconclusive. The proposal of asymmetric valley [25] further throws this debate into limbo, as valleys around minima could be flat on one side but sharp on the other side, which makes it more difficult to define flatness. *This paper thoroughly studies the causes and implications of the unexplained asymmetry phenomenon.*

**Exploiting the Valley Shape of DNNs**. Exploring the valley shape of DNNs could help us better understand the inherent principles of DNNs. The work [43] utilizes 2D surface plots to show that the residual connections in ResNet [27] could prevent the explosion of non-convexity when networks get deep, and [61] attributes the success of BN to its effects of making landscape significantly more smooth. The asymmetric valley [25] provides a sounding explanation for the intriguing phenomenon

in stochastic weight averaging [33]. Additionally, studying the valley shape could also benefit the proposal of effective optimization algorithms, e.g., the entropy-based SGD [6], and the (adaptive) sharpness-aware minimization [16, 41]. Penalizing the gradient could also lead to solutions around flat regions [74]. *We also apply the findings in this paper to the area of model fusion*.

**Model Fusion**. Directly averaging two independent models may encounter a barrier due to training randomness or permutation invariance [4, 22, 2, 51, 24]. However, if the two models are generated from separate training of a common pre-trained model, the model fusion may perform better than individual models, i.e., the model soups [58, 68]. A recent work [70] finds that resolving sign conflicts when merging multiple task-specific models is neccessary, which is most related to our current work. *We will explain the success of model soups based on the relation between the asymmetric valley and the sign consistency of model parameters*. Popular federated learning algorithms also take the parameter averaging process to fuse the individual models updated on isolated clients [54, 52]. A huge challenge is the Non-Independent and Identical Data distributions of data islands (Non-I.I.D. data) [73, 29], which could make local models too diverged to merge. Multiple regularization methods are proposed to align parameters before model fusion [46, 1, 45, 37]. *We propose an effective regularization method that focuses on the sign of parameters, which is inspired by our interesting findings*.

## 3 Basic Notations and Preliminaries

Our major tool is plotting the 1D error curve of DNNs following the formula $\theta_f + \lambda\epsilon$ (Fig. 2 (A)). $\theta_f$ denotes the converged model, and $\epsilon$ denotes a noise vector sampled from a specific distribution. More about the visualization of DNN loss landscape could be found in [43, 49].

### 3.1 Previous Studies: Exploring $\theta_f$ with Fixed $\epsilon$

The previous studies focus on *studying the valley shape under different $\theta_f$*, and aim to *mine the shape's relation to generalization*. To mitigate the influence of parameter scales and make the visualization fairly comparable between different $\theta_f$, [43] proposes the filter normalization method to properly visualize the loss landscape. The processing of the noise is $\epsilon^{i,j} \leftarrow \frac{\epsilon^{i,j}}{\|\epsilon^{i,j}\|}\|\theta_f^{i,j}\|$, where $i$ is the index of layer and $j$ is the index of filter. This way normalizes each filter in the noise $\epsilon$ to have the same norm of the corresponding filter in the converged point $\theta_f$. Further, [41] proposes a proper definition of sharpness (i.e., adaptive sharpness) based on the filter normalization, extending it to all parameters and formally defining: $T_{\theta_f} = \text{diag}\left(\text{concat}\left(\|\mathbf{f}^1\|_2\mathbf{I}_{n_1}, \ldots, \|\mathbf{f}^m\|_2\mathbf{I}_{n_m}, |w^1|, \ldots, |w^q|\right)\right)$, where $\mathbf{f}^j$ with $1 \le j \le m$ denotes the $j$-th convolution filter in $\theta_f$ and $n_j$ is the number of parameters it owns. $w^j$ with $1 \le j \le q$ denotes the $j$-th parameter that is not included in any filters. $\mathbf{I}$ is a vector with all values as one. Then, the adaptive noise $T_{\theta_f}\epsilon$ is utilized to study the sharpness of $\theta_f$.

### 3.2 Our Study: Exploring $\epsilon$ and $\lambda$ with Fixed $\theta_f$

Different from the previous studies, we aim to *explore the valley shape of a fixed $\theta_f$ under different $\epsilon$ and $\lambda$*. First, the direction of 1D interpolation in previous works is limited to the Gaussian noise, while we study the impact of different noise types. Second, setting $\lambda$ positive or negative could obtain a valley with different flatness, i.e., the asymmetric valley [25]. Hence, under the fixed $\theta_f$, we do not need to rectify the noise direction filter-wisely. We take the visualization way used in [31], which only normalizes the noise and then re-scales it to the norm of $\theta_f$, i.e., $\epsilon \leftarrow \frac{\epsilon}{\|\epsilon\|}\|\theta_f\|$. The utilized Norm-Scaled (NS) noise is shown in Fig. 2 (C). Compared with Filter NS noise (Fig. 2 (B)), this way *does not change the direction of the noise and shows the original valley shape along the direction $\epsilon$*. Additionally, to plot 1D error curves in the same figure, we fix $\lambda$ in the range of $[-1, 1]$. Another scale factor $s$ is added to control the visualized width. Overall, we use the following way to plot 1D error curves: $\theta_f + \lambda * s * \frac{\epsilon}{\|\epsilon\|}\|\theta_f\|$, where $\lambda \in [-1, 1]$, and we set $s = 1.0$ by default. The Frobenius norm is used. Notably, we utilize the NS noise by default and use the Filter NS noise when comparing the valley shape under different converged points, e.g., the studying of BN initialization in Sect. 4.2.1.

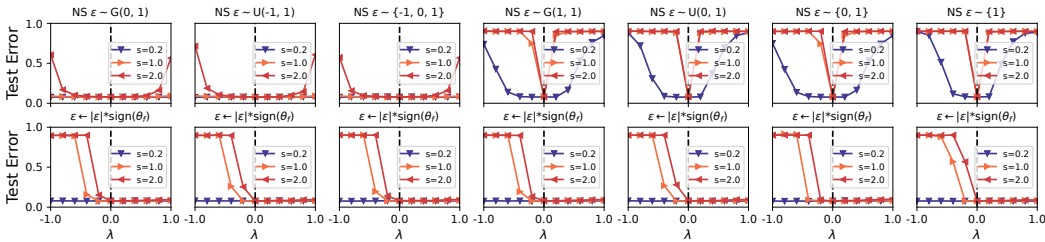

Figure 3: The valleys under 7 common noise types. The second row shows the results of replacing the sign of noise with that of $\theta_f$, leading to asymmetric valleys. (VGG16 with BN on CIFAR10)

## 4 Experimental Findings

This section presents the major findings about factors that respectively impact the noise and converged points. Experimental details and more verification results are in Appendix A and B.

### 4.1 Factors that Affect $\epsilon$

We train VGG16 [64] with BN [32] on CIFAR10 [39] for 200 epochs and obtain the converged model. For the given $\theta_f$, we plot the 1D error curves under the following *7 common noise types*: (1) $G(0,1)$: Gaussian noise with mean as 0 and std as 1; (2) $U(-1,1)$: uniform noise in the range $[-1,1]$; (3) $\{-1,0,1\}$: uniform noise with values only in $-1$, $0$, and $1$; (4) $G(1,1)$: Gaussian noise with mean as 1 and std as 1; (5) $U(0,1)$: uniform noise in the range $[0,1]$; (6) $\{0,1\}$: uniform noise with values only in 0 and 1; (7) $\{1\}$: constant noise with all values as 1. For each plot, we set $s \in \{0.2, 1.0, 2.0\}$ and $\lambda \in [-1,1]$ to show curves under various levels of width. The results are shown in the first row of Fig. 3. We observe that the valley shapes along these 7 noise directions are almost symmetric, except that the last four noise directions show slight asymmetry when $s = 0.2$.

Then, a fantastic idea motivates us to change the sign of the noise. The detail of this motivation is provided in Appendix A.4. Specifically, we use the following method to replace the sign of $\epsilon$ with that of $\theta_f$: $\epsilon \leftarrow |\epsilon| * \text{sign}(\theta_f)$, where $|\cdot|$ returns the absolute value element-wisely and $\text{sign}(\cdot)$ returns 1 or $-1$ based on whether the element is positive or negative. The corresponding results of the 7 common noises become completely asymmetric, which are plotted in the second row of Fig. 3. Furthermore, the valleys all follow the tendency that the positive direction is flat while the negative direction is sharp. Hence, we propose our major finding: *the sign consistency between noise and converged model determines the asymmetry and the valley is flatter along the noise direction with a larger sign consistency*. The finding is formulated as: $L(\theta_f + a\eta) < L(\theta_f - a\eta)$, where $\eta = |\epsilon| * \text{sign}(\theta_f)$ denotes the sign-consistent noise, and $a > 0$ is a constant. $L(\cdot)$ is the loss function, which could be the prediction error or cross-entropy loss. The following three experimental studies could further verify this interesting finding.

**The Manual Construction of Noise Direction**. We element-wise sample the noise $\epsilon$ from $G(0,1)$, and then manually change its elements' sign with a given ratio $r \in \{0.0, 0.1, \ldots, 1.0\}$. For example, $r = 0.5$ means that we sample $50\%$ elements in the noise and change their sign to the same as $\theta_f$. Then, we plot the average test error of the positive interpolations and negative interpolations, i.e., $\mathbb{E}_\lambda[\text{Error}(\theta_f + \lambda\text{NS}(\epsilon))]$ with $\lambda \in [0,1]$ and $\lambda \in [-1,0]$, respectively. Fig. 4 plots the average test errors on two groups of networks and datasets. The test errors of positive and negative directions are nearly equal when $r = 0\%$. As $r$ becomes larger, the average test error of the positive direction monotonically decreases while the negative one increases, implying that the valley shape becomes more and more asymmetric.

**The Investigation of 6 Special Noise Directions**. We then investigate several special noise directions including (1) the initialization before training, i.e., $\epsilon_1 = \theta_0$; (2) the converged model itself, i.e., $\epsilon_2 = \theta_f$; (3) $\epsilon_3 = \text{sign}(\theta_f)$; (4) $\epsilon_4 = \text{sign}(\theta_f - \mu)$; (5) $\epsilon_5 = \text{sgp}(\theta_f)$; (6) $\epsilon_6 = \text{sgp}(\theta_f - \mu)$. Here, $\mu$ denotes the mean value for each parameter group, e.g., the mean value of "conv1.weight". $\text{sgp}(\cdot)$ returns 1 or 0 based on whether the element is positive or not. The visualization results are provided in Fig. 5. The first four directions lead to asymmetry, while the last two do not. First, the elements in $\epsilon_2$

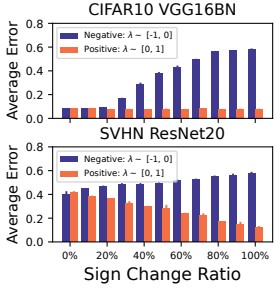

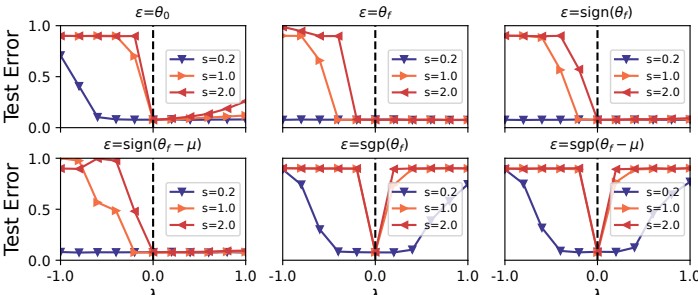

Figure 4: The impacts of manually constructed Gaussian noise with different levels of sign consistency.

Figure 5: The valley shape under 6 special noise types. (VGG16 with BN on CIFAR10)

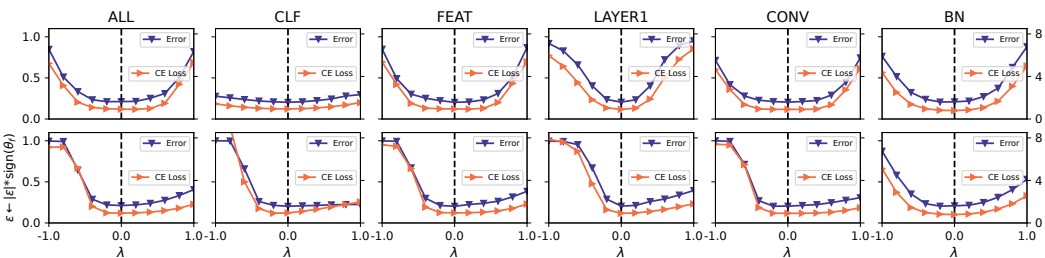

Figure 6: Verification results on ImageNet with pre-trained ResNeXt101.

and $\epsilon_3$ surely have the same sign with $\theta_f$, which leads to an asymmetric valley. Because the mean values of most parameters are near zero, $\epsilon_4$ performs likely as $\epsilon_3$. $\epsilon_5$ and $\epsilon_6$ only have the same sign with the positive parameters in $\theta_f$, and applies zero to negative parameters in $\theta_f$, which shows no asymmetry. The most interesting result is the $\epsilon_1 = \theta_0$, whose elements may be centered around zero according to the Kaiming initialization [26]. However, the BN initialization is asymmetric, which leads to asymmetric curves (Sect. 4.2.1). The results of VGG11 without BN on CIFAR100 [39] show no asymmetry when $\epsilon = \theta_0$ (Appendix B, Fig. 17).

**The Finding Holds for ImageNet and Various Parameter Groups**. We extend the findings to large-scale datasets and apply noise only to specific parameter groups. Specifically, we use the pre-trained models (e.g., ResNeXt101 [69]) downloaded from "torchvision" [1]. Because these models are pre-trained on ImageNet [10], we could directly use them to verify our findings without additional training. Multiple parameter groups are considered as follows: (1) "ALL" denotes the whole parameters; (2) "CLF" denotes the weights in the final classifier layer; (3) "FEAT" denotes the weights in the layers aside from the final classifier; (4) "LAYER1" denotes parameters in the first several blocks; (5) "CONV" denotes all convolution parameters; (6) "BN" denotes all of the BN parameters. As shown in Fig. 6, applying sign-consistent noise could lead to asymmetric valleys. Notably, this holds for both the metrics of CE loss and prediction error.

## 4.2 Factors that Affect $\theta_f$

Then we focus on studying the effects of BN and its initialization, then present the results under various hyperparameters.

---

[1] https://pytorch.org/vision/stable/models.html

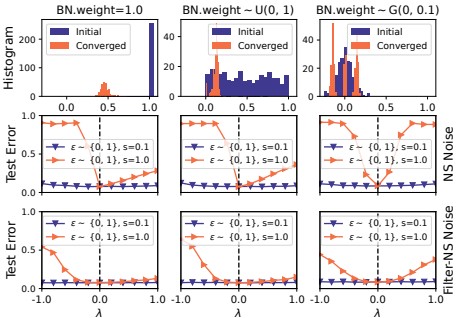
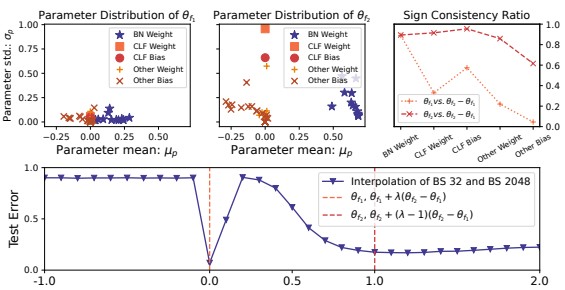

Figure 7: The impact of BN and its initialization on the valley symmetry. (VGG16 with BN on CIFAR10)

Figure 8: The interpolation between two models trained with batch size as 32 and 2048. (VGG16 with BN on CIFAR10)

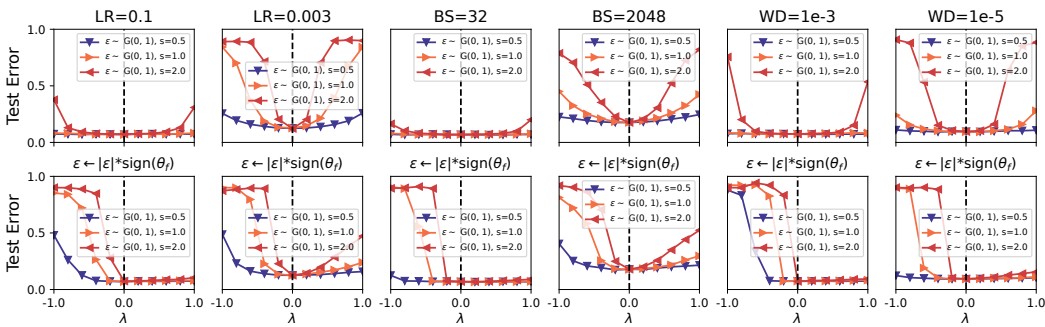

Figure 9: The impact of various hyperparameters on valley symmetry. (VGG16 with BN on CIFAR10)

#### 4.2.1 BN and Initialization

The default initialization method of BN is setting $\mathbf{w}$ as ones and $\mathbf{b}$ as zeros [32, 9]. Hence, the $\mathbf{w}$, i.e., the "BN.weight", may be asymmetric after convergence. We take three ways to initialize the BN weights, including (1) the elements are all ones; (2) the elements are sampled from $U(0,1)$; (3) the values sampled from $G(0,0.1)$. We train three models based on these three types of BN initialization. In Fig. 7, the first row shows the initial and converged parameter distribution of a specific BN weight. The traditional initialization leads to converged BN weights with all positive values, which are nearly centered around $0.2$. The uniform initialization between $0$ and $1$ also leads to positive converged weights. The symmetric Gaussian initialization leads to converged values symmetric around $0$. Then, we plot the valley shapes under the noise direction $\epsilon \in \{0,1\}$. Because this part involves a comparison among different convergence points, we plot the results by both the NS noise and Filter NS noise. As vividly displayed in Fig. 7, the first two initialization ways encounter obvious asymmetry while the Gaussian initialization shows nearly perfect symmetry. If we carefully analyze the sign consistency ratio of them, we could easily explain this phenomenon. The noise direction $\epsilon \in \{0,1\}$ has a larger overlap with the first two initialization methods because the converged BN weights are all positive, while it has a lower overlap with the initialization from $G(0,0.1)$. This implies that *the traditional BN initialization will lead to nearly all positive converged BN weights, which may influence the valley symmetry*.

#### 4.2.2 Hyperparameters

Previously, [38] shows that the batch size could influence the valley width, and further [43] advocates that weight decay could also play a role in the valley width. Different from them, we aim to study whether these hyperparameters influence the valley symmetry. We train VGG16 networks on CIFAR10 with various hyperparameters. We use the SGD optimizer with a momentum value

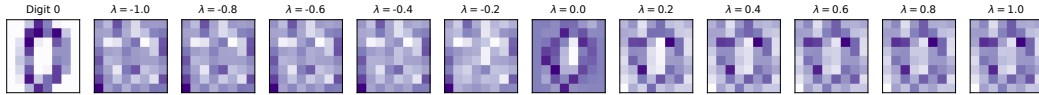

Figure 10: The leftmost shows a digit sample from "sklearn.digits" and others show the pattern of $w + \lambda * \text{sign}(w)$. $\lambda = 0.0$ shows the learned classification weight $w$.

of 0.9. The default learning rate (LR) is 0.03, batch size (BS) is 256, and weight decay (WD) is 0.0005. Then, we will correspondingly set the LR in $\{0.1, 0.003\}$, BS in $\{32, 2048\}$, and WD in $\{0.001, 0.00001\}$. The curves are in Fig. 9. The first row applies the $G(0, 1)$ noise, while the second row changes its sign to the converged models' sign. Obviously, different hyperparameters may lead to valleys with various widths, while the valleys are all symmetric. The asymmetry valleys in the second row again verify the previous findings in Sect. 4.1.

Then, we explore the interpolation between two solutions under different hyperparameters, which are studied in [38, 43, 22]. We take the batch size of 32 and 2048 as an example and denote the converged solution as $\theta_{f_1}$ and $\theta_{f_2}$. The test error curve of $(1 - \lambda)\theta_{f_1} + \lambda\theta_{f_2}$ is plotted, with $\lambda \in [-1, 2]$. Aside from the interpolation curve, we also plot the parameter distributions of $\theta_{f_1}$ and $\theta_{f_2}$. The parameters are divided into five groups, including "BN Weight", "CLF Weight", "CLF Bias", "Other Weight", and "Other Bias". "CLF" denotes the last classification layer, and "Other" denotes other layers aside from the BN layers and the last classification layer. To simplify the figures, we only plot the mean and standard deviation of the parameters, denoted as $\mu_p$ and $\sigma_p$. Fig. 8 shows the parameter distributions and the interpolation curve. The interpolation curve shows that the small batch training (i.e., $\lambda = 0.0$) lies in a sharper and nearly symmetric valley, while the large batch training (i.e., $\lambda = 1.0$) lies in a flatter but asymmetric valley. Small batch training (i.e., $\theta_{f_1}$) leads to parameters with smaller mean and std values. This is because the utilized weight decay is 0.0005, which makes the parameter scale smaller due to longer training [43]. Then, we explain the different results of valley symmetry. The interpolation formula could be re-written as $\theta_{f_1} + \lambda(\theta_{f_2} - \theta_{f_1})$ and $\theta_{f_2} + (\lambda - 1)(\theta_{f_2} - \theta_{f_1})$, which respectively shows the 1D interpolation centered around $\theta_{f_1}$ and $\theta_{f_2}$. If we let $\epsilon = \theta_{f_2} - \theta_{f_1}$, then we could plot the sign consistency ratio (i.e., how many parameters have the same sign) of $\theta_{f_1}$ and $\epsilon$, and $\theta_{f_2}$ and $\epsilon$. The results are in Fig. 8, where we provide the values of the five parameter groups. Obviously, $\theta_{f_2}$ is more consistent in the sign values, which shows a flatter region towards the positive direction. In contrast, the sign consistency ratio of $\theta_{f_1}$ is smaller, which only shows slight asymmetry.

## 5   Theoretical Insights to Explain the Finding

This section provides theoretical insights from the aspects of ReLU activation and softmax function to explain the interesting phenomenon. The forward process of DNNs contains amounts of calculation of $W^T h$, e.g., the fully connected and convolution layer. $W$ denotes the weight matrix and $h$ is a hidden representation. A special case is the final classification layer with the softmax function. Given $h \in R^d$, the ground-truth label $y \in [C]$, and the weight matrix $W \in R^{C \times d}$, the softened probability vector is $p = \text{softmax}(Wh)$. The cross-entropy (CE) loss function is $L(W) = -\log p_y$. The gradient of $w_c$ is $g_{w_c} = -(I\{c = y\} - p_c)h$, with $c \in [C]$ and $I\{\cdot\}$ being the indication function. This implies that the update direction of $W$ lies in the subspace spanned by hidden representations, which also holds for intermediate layers and convolution layers [60]. After adequate training steps, the parameters that activate the ReLU function or correspond to the ground-truth label should correlate well with their inputs. We show a demo classification weight learned on the "sklearn.digits" [2]. Fig. 10 shows the pattern change of $w + \lambda * \text{sign}(w)$ with $\lambda \in [-1.0, 1.0]$. Clearly, the weight under $\lambda = 0.0$ correlates well with the input sample (i.e., Digit 0 in Fig. 10). Setting $\lambda > 0.0$ will almost keep the pattern, while $\lambda < 0.0$ destroys it significantly. That is, $w + \lambda * \text{sign}(w)$ with $\lambda > 0.0$ may keep providing a high score for the target class, while setting $\lambda < 0.0$ may decrease the score.

For the ReLU activation, it also holds that $w + \lambda * \text{sign}(w)$ will have a higher probability of keeping activating the neurons when $\lambda > 0.0$. To simplify the analysis, we assume the learned $w$ equals $a * h + \delta$, where $a$ is a constant and $\delta$ is a random Gaussian vector. Then we could easily verify that

---
[2] https://scikit-learn.org/

$(w + \lambda * \text{sign}(w))^T h$ will have a higher probability of keeping activating neurons under a positive $\lambda$ than the negative one. The details and simulation results are in Appendix D.2. If the neuron outputs are only simply scaled by a factor, it will not affect the relative scores of the final classification. For example, the inequation of $w_1^T h > w_2^T h$ will not change if $h$ is scaled by a positive factor, while it does not hold for $h$ whose values are not activated, i.e., $h = 0$.

Then we provide a further analysis via analyzing the Hessian matrix of the softmax weights. Specifically, the Hessian of $L(W) = -\log p_y$ w.r.t. $W$ is $H = (\text{diag}(p) - pp^T) \otimes hh^T$, where $\otimes$ denotes the Kronecker product. The trace of $H$ is $tr(H) = tr(\text{diag}(p) - pp^T) * tr(hh^T)$. The first part could be calculated as $\sum_c p_c(1 - p_c)$, where $c$ is the class index. According to the above analysis, adding sign-consistent noise to $w_y$ could enlarge the score of $w_y^T h$, which may make the $p_y$ larger and $p_{c \neq y}$ smaller [48]. That is, the predicted probability vector tends to be a one-hot vector when adding sign-consistent noise, and $\sum_c p_c(1 - p_c)$ will be near zero. Hence, the trace of the Hessian matrix is smaller along the sign-consistent direction. Since softmax is convex, and the eigenvalues of the Hessian are all positive, a smaller trace means smaller eigenvalues, which makes the loss curve flatter. The empirical observation can be found in Appendix D.3.

# 6 Applications to Model Fusion

## 6.1 Explaining the Success of Model Soups

Commonly, the interpolation of two independently found DNN solutions may encounter a barrier [22, 65]. Surprisingly, if these two models are updated from the same pre-trained model, then the barrier will disappear, and the linear interpolation brings a positive improvement [58, 68]. The common explanation follows that the pre-trained model may possess less instability when compared to the random initialization models [20]. We guess that the sign consistency ratio may influence the parameter fusion performance of the two models. Perhaps, *the sign of model parameters updated based on the pre-trained model remain nearly unchanged during the process of fine-tuning*.

We experimentally verify our guess on two datasets, i.e., training VGG16BN with *completely random initialization* on CIFAR10, and training ResNet18 with *pre-trained initialization from PyTorch* on Flowers [59]. The datasets are first uniformly split into two partitions, and then two models with corresponding initialization are separately trained or fine-tuned for 50 epochs. The checkpoints in the $\{1, 2, 3, 5, 10, 20, 30, 50\}$-th epoch are stored. For checkpoints in the specific epoch, we denote the two models on the two partitions as $\theta_A$ and $\theta_B$, respectively. The interpolation accuracy of $(1 - \lambda)\theta_A + \lambda\theta_B$ on the test set is plotted in Fig. 11. The interpolation curve of VGG16BN on CIFAR10 indeed encounters a significant barrier, especially when the epoch is larger, e.g., $E = 50$. In contrast, the interpolation surpasses the individual models on Flowers, which is attributed to the pre-trained ResNet18. As an explanation, we calculate the sign consistency ratio between $\theta_A$ and $\theta_I$, $\theta_B$ and $\theta_I$, and $\theta_A$ and $\theta_B$, denoted as "SSR-IA", "SSR-IB", and "SSR-AB", respectively. $\theta_I$ means the initialization model. We also plot the gap of model interpolation and individual models when $\lambda = 0.5$, i.e., $\text{Acc}(0.5\theta_A + 0.5\theta_B) - 0.5(\text{Acc}(\theta_A) + \text{Acc}(\theta_B))$. The right of Fig. 11 clearly shows that the sign consistency ratio could almost perfectly reflect the tendency of the interpolation gap. Notably, the sign consistency ratio of models on Flowers is higher than $0.95$, which means that *fine-tuning the pre-trained model does not change the parameter signs a lot, which facilitates the following parameter interpolation*. For better illustration and fair comparison, we also replace the pre-trained ResNet18 with a random initialized ResNet18, and we find the plots shown in Fig. 11 tend to be like the results of the random initialized VGG16BN. The previous work [70] points out that disagreement on the sign of a given parameter's values across models is a major source of interference during model merging. Although our finding is similar to the previous work, the motivation and specific explanation differs a lot.

## 6.2 Regularizing the Sign Change in Federated Learning

Traditional machine learning models will encounter the challenges posed by "isolated data islands", e.g., the Non-I.I.D. data [54, 29]. FedAvg [54], as the most standard federated learning (FL) method, utilizes the parameter server architecture [44], and fuses collaborative models from local nodes without centralizing users' data. Specifically, the local clients receive the global model from the server and update it respectively on their devices using private data, and the server periodically

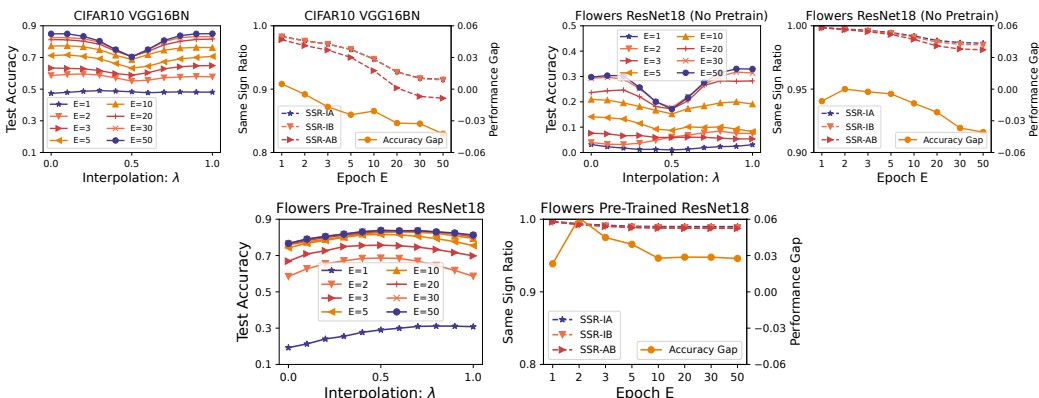

Figure 11: The models fine-tuned from a pre-trained model have a higher sign consistency ratio. The top two sub-figures do not utilize pre-trained models, while the last one utilizes pre-trained ResNet18.

Table 1: Aggregation performance comparisons of FedSign with several popular FL algorithms.

|          | Dir. $\alpha$ | FedAvg | FedProx | MOON | FedDyn | FedPAN | FedSign |
|----------|------|--------------|---------|-------|--------|--------|----------------|
| CIFAR-10 | 10.0 | $81.53 \pm 0.17$ | 81.84 | 82.44 | 80.45 | 81.92 | **82.59** $\pm 0.09$ |
|          | 1.0  | $80.54 \pm 0.11$ | 80.42 | 80.12 | 79.78 | 80.30 | **80.76** $\pm 0.14$ |
|          | 0.5  | $77.69 \pm 0.21$ | 78.12 | 76.77 | 78.02 | 77.78 | **78.41** $\pm 0.35$ |
| CINIC-10 | 10.0 | $75.74 \pm 0.24$ | 76.58 | 76.25 | 76.37 | 76.84 | **77.05** $\pm 0.14$ |
|          | 1.0  | $72.19 \pm 0.15$ | 72.25 | 72.06 | 73.12 | 73.46 | **74.59** $\pm 0.20$ |
|          | 0.5  | $68.24 \pm 0.32$ | 69.11 | 68.55 | 69.01 | 70.14 | **70.63** $\pm 0.16$ |

*averages* these models for multiple communication rounds. That is, FedAvg takes the simple parameter averaging to fuse local models. However, due to the Non-I.I.D. data and permutation invariance of DNNs [51, 67, 72], the local models could become too diverged to effectively merge. Numerous efforts are paid to regularize the local models so as not to go too far away from the global model, such as the proximal term proposed by FedProx [46], the contrastive regularization proposed by MOON [45], the dynamic regularization of FedDyn [1], and the position-aware neurons proposed by FedPAN [51]. Inspired by our finding, we propose to regularize the sign change when updating local models, i.e.,

$$\mathcal{L}^k = \mathcal{L}_{\text{ce}}^k - \gamma \left( \text{sgp}(\theta_t)\sigma(\theta_t^k) + \text{sgp}(-\theta_t)\sigma(-\theta_t^k) \right), \tag{1}$$

where $t$ denotes the communication round and $k$ is the index of client. $\mathcal{L}_{\text{ce}}^k$ is the common cross-entropy loss of the $k$-th client, and $\theta_t$ is the global model received from the server. $\theta_t^k \leftarrow \theta_t$ is the local model to be updated, and the loss could regularize the sign of $\theta_t^k$ to be close to that of $\theta_t$. Because obtaining the sign of parameters is not a continuous function, we therefore apply a sigmoid function to parameters as an approximation. We name this method FedSign and list its pseudo-code in the Appendix C. Compared with other regularization methods, our proposed FedSign is well-motivated because of our finding that interpolating sign-consistent models may lead to a flatter loss region.

Experimental studies are verified on CIFAR10 [39] and CINIC10 [8] that are commonly utilized in previous works [72, 51]. Decentralizing the training data of these datasets by a Dirichlet distribution could simulate the Non-I.I.D. scenes as in real-world FL. The Dirichlet alpha $\alpha$ is utilized to control the Non-I.I.D. level, with a smaller $\alpha$ representing a more rigorous heterogeneity between clients' data. We set $\alpha \in \{10.0, 1.0, 0.5\}$ respectively. The number of clients is 100, and the total communication round is 200. During each round, a random set of 10% clients participate in FL and every client takes 5 epochs update on their individual data. After all communication rounds, we evaluate the model performance on a global test set on the server (i.e., the original test set of corresponding datasets). We select our hyperparameter $\gamma$ from $\{0.001, 0.01, 0.1\}$ and report the best results. For $\alpha = 10.0$, i.e., a relatively I.I.D. scene, a smaller $\gamma$ is better. In contrast, $\gamma = 0.1$ or $\gamma = 0.01$ will be more proper for $\alpha = 0.5$. The performance comparison results are listed in Tab. 1. For FeaAvg and our proposed FedSign, we rerun the experimental studies five times and list the standard deviation of accuracies,

showing that the accuracy doesn't fluctuate very much. FedSign could surpass the compared methods, which shows the positive effects of regularizing the sign change in FL.

# 7 Limitations and Future Works

Although we provide theoretical insights to explain the interesting phenomenon, no formal proofs are provided to show the conditions and scopes that lead to asymmetric valleys. Additionally, this phenomenon is only investigated in the image classification tasks. Future research includes providing formal theoretical foundations for our findings and verifying them on more tasks. According to the analysis in Sect. 5, we advocate that this phenomenon is more likely to be applicable to DNNs that contain both the ReLU and softmax.

# 8 Conclusion

We explore and exploit the asymmetric valley of DNNs via numerous experimental studies and theoretical analyses. We systematically examine various factors influencing valley symmetry, highlighting the significant role of sign consistency between noise direction and the converged model. The findings offer valuable insights into practical implications, enhancing the understanding of model fusion. A novel regularization method is proposed for better model averaging in federated learning.

# Acknowledgements

This work is partially supported by National Science and Technology Major Project (2022ZD0114805), NSFC (62376118, 62006112, 62250069, 61921006), Collaborative Innovation Center of Novel Software Technology and Industrialization. Professor De-Chuan Zhan is the corresponding author.

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

# A  Experimental Details

In this section, we list the datasets, networks, and training details that are utilized in the body. Finally, we provide the details that motivate us to change the sign of the noise.

## A.1  Dataset

The utilized datasets include "sklearn.digits" [3], SVHN [56], CIFAR10/100 [39], CINIC10 [8], Flowers [59], Food101 [5], and ImageNet [10]. We detail these datasets as follows.

- **"sklearn.digits"** contains 1797 samples of 10 digits, with each sample being a $8 \times 8$ image. We use this to provide a simple code demo and provide theoretical verification as in Appendix D.3.

- **SVHN** [56] is the Street View House Number dataset which contains 10 numbers to classify. The raw set contains 73,257 samples for training and 26,032 samples for evaluation. The image size is $32 \times 32$.

- **CIFAR10** and **CIFAR100** [39] are subsets of the Tiny Images dataset and respectively have 10/100 classes to classify. They consist of 50,000 training images and 10,000 test images. The image size is $32 \times 32$.

- **CINIC10** [8] is a combination of CIFAR10 and ImageNet [40], which contains 10 classes. It contains 90,000 samples for training, validation, and testing, respectively. We do not use the validation set. The image size is $32 \times 32$.

- **Flowers** [59] consists of 102 fine-grained flower categories, where we select 2,000 images for training and 2,000 images for testing.

- **Food101** [5] consists of 101 food categories, with 101, 000 images. For each class, 250 manually reviewed test images are provided as well as 750 training images. All images were rescaled to have a maximum side length of 512 pixels.

- **ImageNet** [10] consists 1000 image categories for classification. This dataset is utilized to pre-train models as listed in "torchvision". Due to the large amounts of data, we only select 5,000 images in the "val" partition to plot the curves in corresponding figures.

## A.2  Network Details

We utilize VGG [64], ResNet [27], ResNeXt [69], AlexNet [40], ViT [13] in this paper. We detail their architectures as follows:

- **VGG** contains a series of networks with various layers. The paper of VGG [64] presents VGG11, VGG13, VGG16, and VGG19. We follow their architectures and report the configuration of VGG11 as an example: 64, M, 128, M, 256, 256, M, 512, 512, M, 512, 512, M. "M" denotes the max-pooling layer. VGG11 contains 8 convolution blocks and three fully-connected layers in [64]. The VGG architecture could use BatchNorm [32] or not, which is clearly declared in the body. If BN is used, it will be added after each convolution layer and before the ReLU activation function.

- **ResNet** introduces residual connections to plain neural networks. We take the CIFAR versions used in the paper [27] for CIFAR10/100 and CINIC10, i.e., ResNet20 with the basic block. For Flowers, we use pre-trained ResNet18 from PyTorch. ResNet commonly uses BatchNorm [32], which is added before ReLU activation.

- **AlexNet** [40] consists of five convolutional layers followed by three fully connected layers.

- **ResNeXt** [69] introduces the group convolution to ResNet and we utilize the pre-trained version downloaded from "torchvision".

- **ViT** [13] follows the transformer architecture for image classification tasks. It divides an image into a sequence of fixed-size patches, processes these patches linearly, and then feeds them into a transformer encoder to capture the global context of the image. In our paper, we

---

[3]`https://scikit-learn.org/stable/modules/generated/sklearn.datasets.load_digits.html`

take 12 layers in the transformer encoder and use 8 heads in each multi-head self-attention block. We set the embedding dimension as 128 to reduce the computation burden.

## A.3 Training Details

We provide the training details for obtaining converged models. We investigate the following pairs of datasets and networks. For the series of Fig. 3 and Fig. 5, we train VGG16BN on CIFAR10, ResNet20 on SVHN, VGG11 with no BN on CIFAR100, AlexNet on Food101, and ViT on Food101. For VGG16BN, ResNet20, VGG11 with no BN, and AlexNet, we use the SGD optimizer with a momentum of 0.9. We set the learning rate as 0.03. We use a cosine annealing way to decay the learning rate across 200 training epochs. The default weight decay is 0.0005, and the default batch size is 256. For training ViT, we use the AdamW optimizer with a learning rate of 0.0001. The batch size is 256, and the weight decay is 0.0005. We also take the cosine annealing way to decay the learning rate.

The equation of the BN layer is $\mathbf{X} = \mathbf{w}\frac{\mathbf{X}-\mathbf{m_X}}{\sqrt{\mathbf{v_X}+\eta}} + \mathbf{b}$, where $\mathbf{X} \in \mathcal{R}^{C \times d}$ denotes the feature map with $C$ channels, and $\mathbf{m_X} \in \mathcal{R}^C$ and $\mathbf{v_X} \in \mathcal{R}^C$ are channel-wise mean and variance values of the feature map. In practice, $\mathbf{w}, \mathbf{b} \in \mathcal{R}^C$ are learnable parameters, while $\mathbf{m_X}$ and $\mathbf{v_X}$ are running statistics that are calculated during the forward pass. When interpolating $\theta_f + \lambda\epsilon$ with BN layers, we should clear these running statistics after interpolating model parameters and feed the interpolated model to the dataset for another forward pass to calculate proper data distributions. The forward-again process is also utilized in previous works [33, 68].

For Fig. 7, we only change the initialization method of BN layers and keep the other hyperparameters not changed. For the series of Fig. 8, we only change a specific hyperparameter including the learning rate, batch size, or weight decay. Specifically, the learning rate is varied in $\{0.1, 0.003\}$, and the batch size is varied in $\{32, 2048\}$, and the weight decay is varied in $\{0.001, 0.00001\}$.

## A.4 Motivation of Changing the Sign of Noise

The asymmetric valley is initially proposed by [25], while it does not propose the inherent principles behind the phenomenon. It only points out that adding asymmetric noise (e.g., $\epsilon \sim \{0, 1\}$) to DNNs may result in an asymmetric valley. The symmetric noise around zero may not show such patterns (e.g., $\epsilon \sim \{-1, 0, 1\}$). This inspires us to plot valleys along three types of symmetric noise directions and four types of asymmetric noise directions, and the results are shown in Fig. 3. Indeed, the last four types of noise are not symmetric around zero and show slight asymmetric valleys. However, this is not so obvious. Notably, the utilized network in Fig. 3 has BN layers, i.e., VGG16 with BN.

Then, we try to apply $\epsilon \sim \{0, 1\}$ to DNNs without BN, i.e., VGG11 without BN. The valleys become symmetric as shown in the first row of Fig. 13. Hence, this makes us consider the effect of BN layers. Fortunately, we find that the traditional BN initialization will initialize the values in "BN.weight" as 1, and the converged BN weights are all positive. The initial findings could be summarized as:

- If DNNs have BN, and the parameters are perturbed by noise with symmetric values around zero, the valleys are symmetric. This is shown as the top first three plots in Fig. 3.

- If DNNs have BN, and the parameters are perturbed by noise with asymmetric values around zero, the valleys are slightly asymmetric. This is shown as the top last four plots in Fig. 3.

- If DNNs do not have BN, and the parameters are perturbed by noise with symmetric values around zero, the valleys are symmetric. This is shown as the top first three plots in Fig. 13.

- If DNNs do not have BN, and the parameters are perturbed by noise with asymmetric values around zero, the valleys are symmetric. This is shown as the top last four plots in Fig. 13.

That is, only the second case shows slightly asymmetric valleys, where the noise (e.g., $\epsilon \in \{0, 1\}$) has a large sign consistency with the BN weights (e.g., $> 0.0$). As shown in Sect. 4.2.1, if we replace the initialization of BN weights with a random Gaussian initialization, then the plotted valley becomes symmetric because the converged BN weights are symmetric around zero again (Fig. 7).

That is, the converged "BN.weight" are all positive values under the common BN initialization. If we perturb them by asymmetric noise (e.g., $\epsilon \in \{0, 1\}$, $\epsilon \in G(1, 1)$, $\epsilon \in U(0, 1)$, or $\epsilon \in \{1\}$) that has

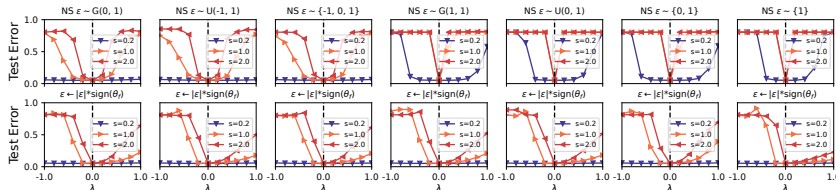

Figure 12: The valleys under 7 common noise types. The second row shows the results of replacing the sign of noise with that of $\theta_f$, leading to asymmetric valleys. (ResNet20 on SVHN)

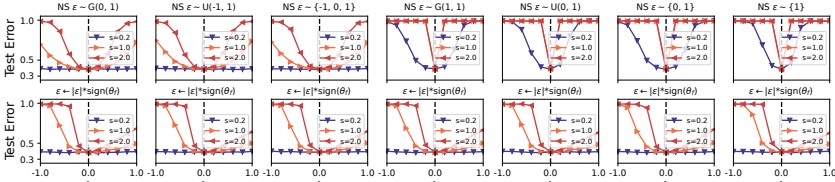

Figure 13: The valleys under 7 common noise types. The second row shows the results of replacing the sign of noise with that of $\theta_f$, leading to asymmetric valleys. (VGG11 without BN on CIFAR100)

a larger sign consistency with "BN.weight", the plotted valleys are asymmetric. This provides an explanation for the asymmetric valley found by [25].

From the above summary, we guess that the sign consistency between parameters and the noise may lead to asymmetric valleys. Hence, a fantastic idea motivates us to change the sign of the noise to the same as $\theta$, which leads to obvious asymmetric valleys (the bottom row of Fig. 3 and Fig. 13).

# B    More Experimental Studies

In this section, we provide more experimental verification results to make our work more solid. We list the supplemented results by the corresponding experimental studies in the body.

## B.1    More Results as Fig. 3

Fig. 3 shows the valley shape under 7 common noise types. The results are plotted via training VGG16 with BN on CIFAR10. We list more results to verify the finding is common across various networks and architectures, including: (1) ResNet20 on SVHN (Fig. 12); (2) VGG11 without BN on CIFAR100 (Fig. 13). Obviously, these results are indeed similar, which verifies again that the sign consistency ratio matters a lot in the valley symmetry.

We also extend the findings to the large-scale dataset and popular network architectures including: (1) AlexNet on Food101 (Fig. 14); (2) ViT on Food101 (Fig. 15). Applying noise with a higher sign

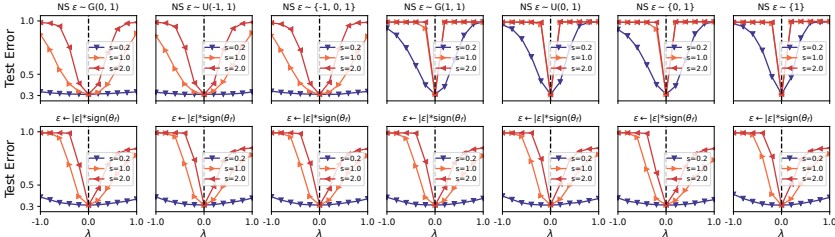

Figure 14: The valleys under 7 common noise types. The second row shows the results of replacing the sign of noise with that of $\theta_f$, leading to asymmetric valleys. (AlexNet on Food101)

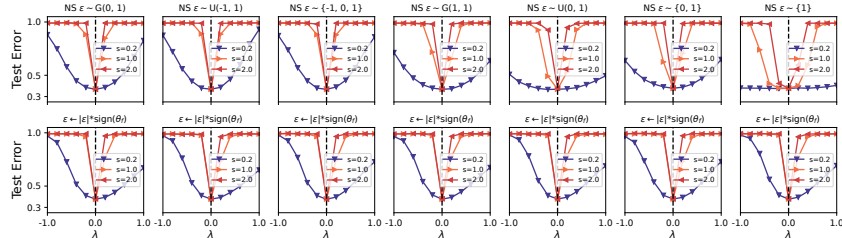

Figure 15: The valleys under 7 common noise types. The second row shows the results of replacing the sign of noise with that of $\theta_f$, leading to asymmetric valleys. (ViT on Food101)

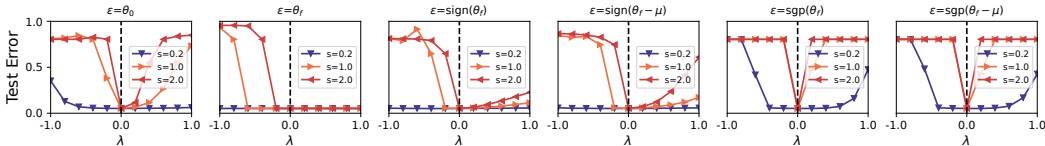

Figure 16: The valley shape under 6 special noise types. (ResNet20 on SVHN)

consistency also leads to asymmetric valleys. However, the large models are relatively stable to some extent, and the asymmetry is not as obvious as the results on the previous datasets and networks.

## B.2 More Results as Fig. 5

Fig. 5 shows the valley shape under 6 special noise types. The results are plotted via training VGG16 with BN on CIFAR10. We list more results to verify the finding is nearly common across various networks and architectures, including: (1) ResNet20 on SVHN (Fig. 16); (2) VGG11 without BN on CIFAR100 (Fig. 17). Obviously, these results are indeed similar. An exceptional case is the first sub-figure in Fig. 17, i.e., using the initialization as noise leads to a symmetric valley, while the valley in Fig. 5 and Fig. 16 is asymmetric. This is because of the initialization of BN parameters, where the latter two utilize BN weights as all ones. However, Fig. 17 takes VGG11 without BN layers, which shows no asymmetry.

We also extend the findings to the large-scale dataset and popular network architectures including: (1) AlexNet on Food101 (Fig. 18); (2) ViT on Food101 (Fig. 19). Similar to the previous results, although the asymmetry is not as obvious as the results on the previous datasets and networks, the curves still show slight asymmetry when the sign consistency is high.

## B.3 More Results as Fig. 6

Fig. 6 shows the valley shape investigated on ImageNet with pre-trained ResNeXt101. We also provide similar results with the pre-trained ResNet50 as in Fig. 20.

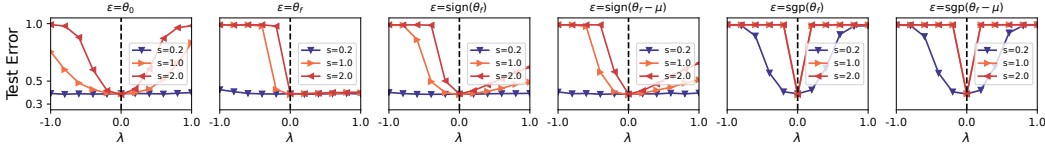

Figure 17: The valley shape under 6 special noise types. (VGG11 without BN on CIFAR100)

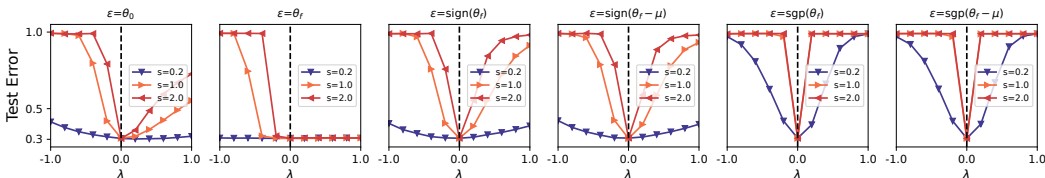

Figure 18: The valley shape under 6 special noise types. (AlexNet on Food101)

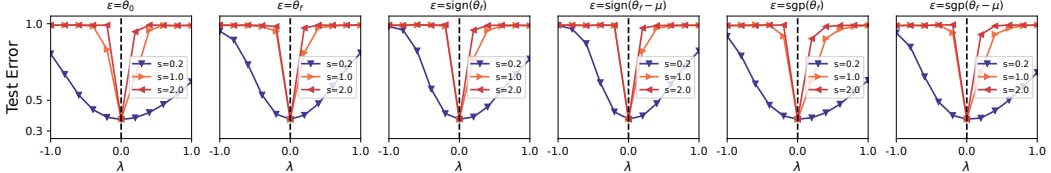

Figure 19: The valley shape under 6 special noise types. (ViT on Food101)

## B.4 More Results as Fig. 7

Fig. 7 shows the impact of BN and its initialization on the valley symmetry. The results are plotted via training VGG16 with BN on CIFAR10. We list more results to verify the finding is common across various networks and architectures, including (1) ResNet20 on SVHN (Fig. 21); (2) ResNet20 on CIFAR100 (Fig. 22). Obviously, these results are indeed similar, which shows that the original BN initialization may lead to an asymmetric valley. Replacing the original initialization with the symmetric Gaussian distribution could lead to symmetric valleys.

## B.5 More Results as Fig. 9

Fig. 9 shows the impact of various hyperparameters on valley symmetry. The results are plotted via training VGG16 with BN on CIFAR10. We list more results to verify the finding is common across various architectures, i.e., ResNet20 on CIFAR10 (Fig. 23). This also verifies that the hyperparameters have less impact on the valley symmetry.

## B.6 More Results as Fig. 8

Fig. 8 studies the interpolation between two models that trained under different hyperparameters, e.g., learning rate (LR), batch size (BS), and weight decay (WD). The body only shows the impact of batch size on training VGG16BN. Then, we list more results to verify the finding is common across various architectures, including: (1) the impact of learning rate on training VGG16BN on CIFAR10 (Fig. 24); (2) the impact of weight decay on training VGG16BN on CIFAR10 (Fig. 25); (3)

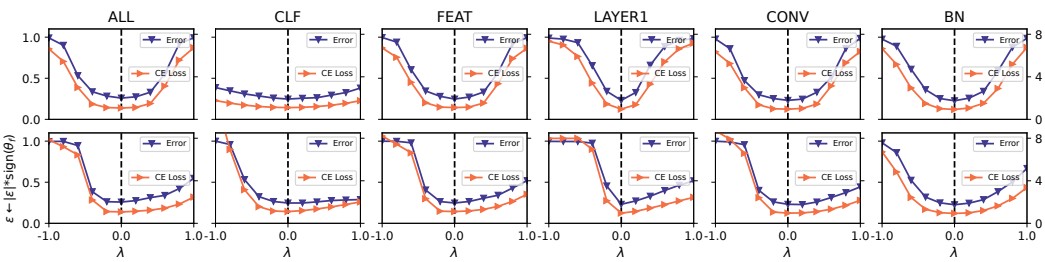

Figure 20: Verification results on ImageNet with pre-trained ResNet50.

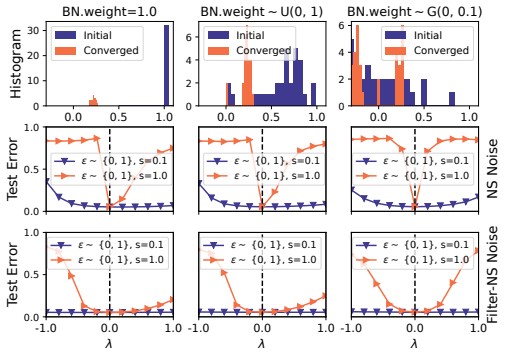
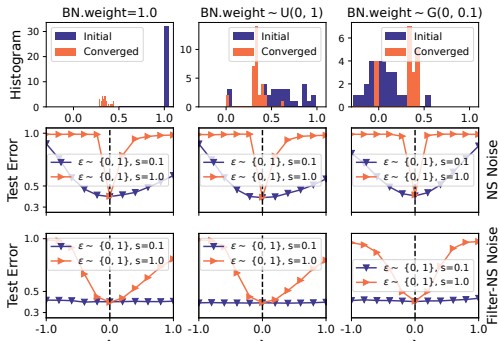

Figure 21: The impact of BN and its initialization on the valley symmetry. (ResNet20 on SVHN)

Figure 22: The impact of BN and its initialization on the valley symmetry. (ResNet20 on CIFAR100)

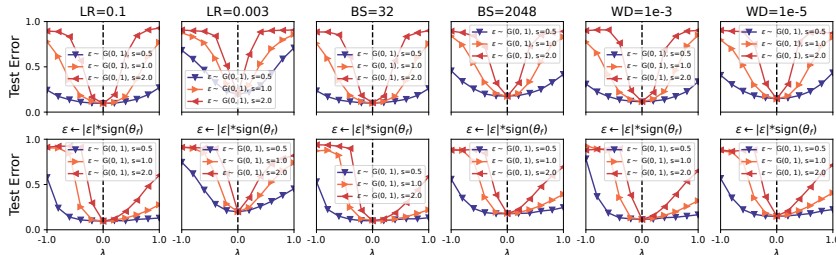

Figure 23: The impact of various hyperparameters on valley symmetry. (ResNet20 on CIFAR10)

the impact of batch size on training ResNet20 on CIFAR10 (Fig. 26). From these figures, we could again observe that the parameter scale influences the valley width, while the sign consistency ratio matters to the valley symmetry.

Additionally, we could further obtain the following conclusion: a larger learning rate (e.g., 0.1), a smaller batch size (e.g., 32), or a larger weight decay (e.g., 0.001) could lead to better performances, which are shown in these figures with a lower test error when compared the opposite hyperparameter. However, their parameter scales are relatively smaller than opposite ones, making their valley width sharper. And commonly, a larger parameter scale of $\theta_{f_1}$ will let the sign of $\theta_{f_1} - \theta_{f_2}$ conform to $\theta_{f_1}$ more, which leads to a flatter region along the positive direction of $\theta_{f_2}$.

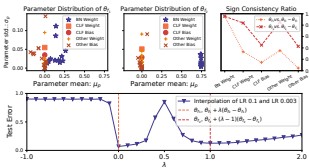
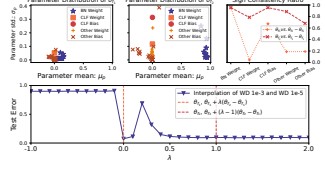
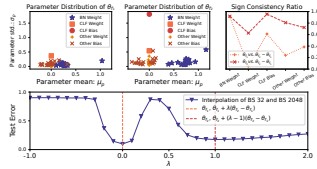

Figure 24: The interpolation between two models trained with learning rate as 0.1 and 0.003. (VGG16 with BN on CIFAR10)

Figure 25: The interpolation between two models trained with weight decay as 0.001 and 0.00001. (VGG16 with BN on CIFAR10)

Figure 26: The interpolation between two models trained with batch size as 32 and 2048. (ResNet20 on CIFAR10)

---

**Algorithm 1** FedSign

---

**ServerProcedure**:
 1: **for** global round $t = 0, 1, 2, \ldots, T$ **do**
 2:    $S_t \leftarrow$ sample $\max(Q \cdot K, 1)$ clients
 3:    **for** $k \in S_t$ **do**
 4:       $\hat{\theta}_t^k \leftarrow$ ClientProcedure$(k, \theta_t)$
 5:    **end for**
 6:    $\theta_{t+1} \leftarrow \sum_{k=1}^{|S_t|} \frac{1}{|S_t|} \hat{\theta}_t^k$
 7: **end for**

**ClientProcedure**$(k, \theta_t)$:
 1: $\theta_t^k \leftarrow \theta_t$
 2: **for** local epoch $e = 1, 2, \ldots, E$ **do**
 3:    **for** each batch with $B$ samples from $\mathcal{D}^k$ **do**
 4:       Calculate the loss as in Eq. 1, update $\theta_t$ using, e.g., SGD with momentum
 5:    **end for**
 6: **end for**
 7: **Return**: the updated model $\hat{\theta}_t^k$

---

## C Pseudo Code of FedSign

We provide a pseudo code of the application to federated learning, i.e., FedSign. Our proposed FedSign takes a novel regularization method to limit the change of local models' signs, whose goal is for better model fusion on the server. The pseudo-code is listed as in Algo. 1. Here, $T$ denotes the number of communication rounds, $K$ is the total number of clients, $Q$ is a client participating ratio in each round, $E$ is the local update epochs of participated clients, $B$ denotes the batch size, and $\mathcal{D}^k$ denotes the private data of the $k$-th client. In our experimental studies, we set $T = 200$, $K = 100$, $Q = 10\%$, $E = 5$, and $B = 64$. After all communication rounds, we obtain the final aggregated model $\theta_{T+1}$, and then we test its accuracy on the global test set to evaluate the aggregation performance of FedSign.

## D Detailed Theoretical Analysis and Verification

Our major finding is formulated as: $L(\theta_f + a\eta) < L(\theta_f - a\eta)$, where $\eta = |\epsilon| * \text{sign}(\theta_f)$ denotes the sign-consistent noise, and $a > 0$ is a constant. $L(\cdot)$ is the loss function. This finding holds for several settings, including (1) the cases when $L(\cdot)$ is the prediction error or cross-entropy loss, which are verified in Fig. 6 and Fig. 20; (2) the cases of applying noise to whole parameters, only to softmax classification layer, or other layers, which are verified in Fig. 6 and Fig. 20.

We present theoretical analysis from several possible aspects and finally attribute this finding to the properties of ReLU activation and softmax classification.

### D.1 Gradient Analysis

First, we guess that the $\text{sign}(\theta_f)$ may have a correlation to the gradient. Specifically, if $\epsilon = \nabla_{\theta_f} \mathcal{L}(\theta_f)$, then given a very smaller $\lambda$, we may have the following relationship:

$$\mathcal{L}(\theta_f + \lambda \nabla_{\theta_f} \mathcal{L}) \geq \mathcal{L}(\theta_f) \geq \mathcal{L}(\theta_f - \lambda \nabla_{\theta_f} \mathcal{L}), \tag{2}$$

where the first and third terms denote the gradient ascent step and gradient descent step, respectively. If we set $\epsilon = \nabla_{\theta_f} \mathcal{L}(\theta_f)$ and $\lambda \in [-1, 1]$, the plotted curve may be asymmetric. However, we frustratingly find that $\text{sign}(\theta_f)$ are almost orthogonal to the $\nabla_{\theta_f} \mathcal{L}$. On one hand, the converged model $\theta_f$ has very small gradient values, and elements in $\nabla_{\theta_f} \mathcal{L}$ are nearly zero. On the other hand, adding $\text{sign}(\theta_f)$ to $\theta_f$ cannot obtain a lower error than $\theta_f$ itself, which is not the same as the gradient descent direction. This implies that the inherent properties of them are not the same. Hence, we try other possible explanations.

## D.2 ReLU Activation

The asymmetry may be related to the inherent asymmetry of the ReLU activation function. According to the analysis in Sect. 5, the gradient of parameters lies in the subspace spanned by the corresponding inputs. Hence, the converged parameters may correlate with their inputs to some extent. Fig. 10 shows the pattern of learned classification weight on the "sklearn.digits". The learned classification weight $w$ shows a pattern of "0" as shown in the figure. However, *it is hard for us to provide a concrete expression about the correlation of weight with its corresponding inputs, even for the simplest softmax classification layer*. Hence, we assume that the learned $w$ equals to $a * h + \delta$, where $a$ is a constant and $\delta$ is a random Gaussian vector. $h$ denotes the hidden representation. And then we simulate the distribution of $(w + \lambda * \text{sign}(w))^T h$ by the following Python code 1. Specifically, we set $h \in R^d$ as a vector sampled from the distribution of $G(0, 1)$, and set $a = 0.1$. We take $\delta$ as a Gaussian vector sampled from the distribution of $G(0, 1)$. Then we sample $N = 10000$ groups of $\delta$ and plot the distribution of $(w + \lambda * \text{sign}(w))^T h$ under $\lambda \in \{-1.0, -0.5, 0.0, 0.5, 1.0\}$. The distributions are shown in Fig. 27. Obviously, with an increasing $\lambda$, the distribution is shifted right. In other words, a negative $\lambda$ may decrease the value of $(w + \lambda * \text{sign}(w))^T h$. In ReLU activation, the negative values are not activated, making the loss error change a lot.

Listing 1: Simulate ReLU

```python
import numpy as np
from matplotlib import pyplot as plt

d = 512
a = 0.1
h = np.random.randn(d)
lambs = np.linspace(-1.0, 1.0, 5)

plt.figure(figsize=(10, 3))
for lamb in lambs:
    vs = []
    for _ in range(10000):
        w = np.random.randn(d) + 0.1 * h
        vs.append(np.dot(w + lamb * (2.0 * (w > 0.0) - 1.0), h))
    plt.hist(vs, bins=500)
plt.legend(["lamb={}".format(lamb) for lamb in lambs])
plt.show()
```

Aside from the simulation demo, we also provide the hidden activations for ResNet20 trained on SVHN. The results are shown in Fig. 28. We plot the activation confusion matrix of $\theta_f + \lambda * |\epsilon| \text{sign}(\theta_f)$ with $\lambda \in [-1.0, 1.0]$. For each $\lambda$, we obtain the hidden features extracted by $\theta_f + \lambda * |\epsilon| \text{sign}(\theta_f)$. We then compare the features with the original features extracted by $\theta_f$ and plot the activation confusion matrix. The sum of diagonal values represents the outputs that the original model and the interpolated model commonly activate or do not activate. The value in "[]" shows the sum of diagonal values. Obviously, the value is larger when $\lambda = a$ than that of $\lambda = -a$, with $a \in \{0.2, 0.4, 0.6, 0.8, 1.0\}$.

To conclude, *for the ReLU activation, adding sign-consistent noise to parameters will have a higher probability of activating the neurons*. If the neuron outputs are only simply scaled by a factor, it will not affect the relative scores of the final classification. For example, the inequation of $w_1^T h > w_2^T h$ will not change if $h$ is scaled by a positive factor, while it does not hold for $h$ whose values are not activated, i.e., $h = 0$.

## D.3 Softmax Function

Given $h \in R^d$, the ground-truth label $y \in [C]$, and the weight matrix $W \in R^{C \times d}$, the softened probability vector is $p = \text{softmax}(Wh)$. The cross-entropy (CE) loss function is $L(W) = -\log p_y$. The gradient of $w_c$ is $g_{w_c} = -(I\{c = y\} - p_c)h$, with $c \in [C]$ and $I\{\cdot\}$ being the indication function. Specifically, the Hessian of $L(W) = -\log p_y$ w.r.t. $W$ is $H = (\text{diag}(p) - pp^T) \otimes hh^T$, where $\otimes$ denotes the Kronecker product. The trace of $H$ is $tr(H) = tr(\text{diag}(p) - pp^T) * tr(hh^T)$. The first part could be calculated as $\sum_c p_c(1 - p_c)$, where $c$ is the class index.

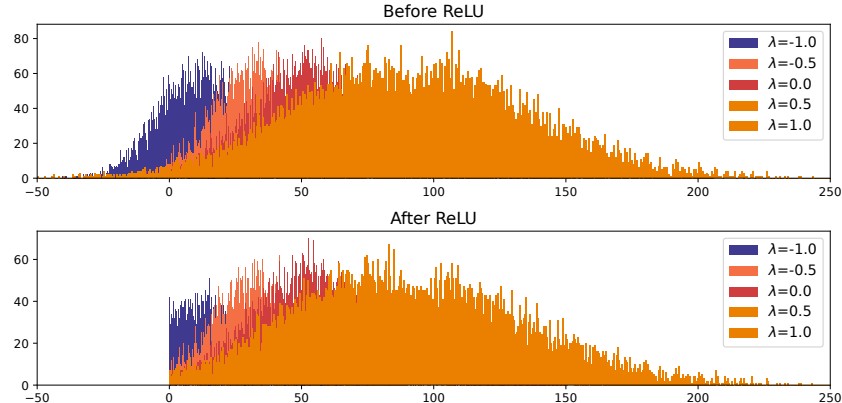

Figure 27: The distribution of $(w + \lambda * \text{sign}(w))^T h$ with $w = 0.1 * h + \delta$. $h$ and $\delta$ are sampled from $G(0,1)$.

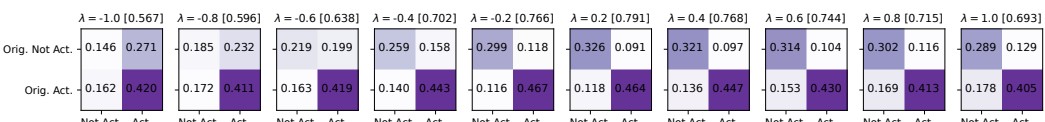

Figure 28: The activation confusion matrix of $\theta_f + \lambda * |\epsilon|\text{sign}(\theta_f)$ with $\lambda \in [-1.0, 1.0]$.

We use the "sklearn.digits" dataset and train a softmax classification weight via the "LogisticRegression" classifier. The demo code is listed in Python code 2. When the training finishes, we obtain the converged weight $W$. Then, we perturb it via the Gaussian noise $\epsilon$ and the sign-consistent Gaussian noise $|\epsilon| * \text{sign}(W)$, respectively. Given a $\lambda \in [-1, 1]$, we calculate several metrics: (1) the test error; (2) the cross-entropy loss; (3) the average trace of $P_\lambda = \text{diag}(p) - pp^T$, i.e., $E_x[tr(P_\lambda)]$; (4) the trace of the Hessian matrix, i.e., $tr(H_\lambda)$; (5) the coefficient of the first-order approximation of $\mathcal{L}$ w.r.p. $\lambda$, i.e., $\epsilon^T g_\lambda$, where $g_\lambda$ denotes the gradient w.r.p. $W + \lambda|\epsilon| * \text{sign}(W)$; (6) the coefficient of the second-order approximation of $\mathcal{L}$ w.r.p. $\lambda$, i.e., $\epsilon^T H_\lambda \epsilon$, where $H_\lambda$ denotes the Hessian matrix w.r.p. $W + \lambda|\epsilon| * \text{sign}(W)$. The above metrics are shown in Fig. 29. The test error and cross-entropy loss in the second row show asymmetry, because the second row takes the sign-consistent noise. More specifically, with a positive $\lambda = a$, the average trace of $P_\lambda$ is smaller than under a negative $\lambda = -a$. Therefore, the trace of $H_{\lambda=a}$ is smaller than that of $H_{\lambda=-a}$. A smaller trace of Hessian means a flatter loss region.

The analysis from ReLU and softmax explain the observed phenomenon in this paper, i.e., the sign-consistent noise leads to valley asymmetry and the positive direction is flatter. However, the theoretical insights are not formal proofs, which are future works.

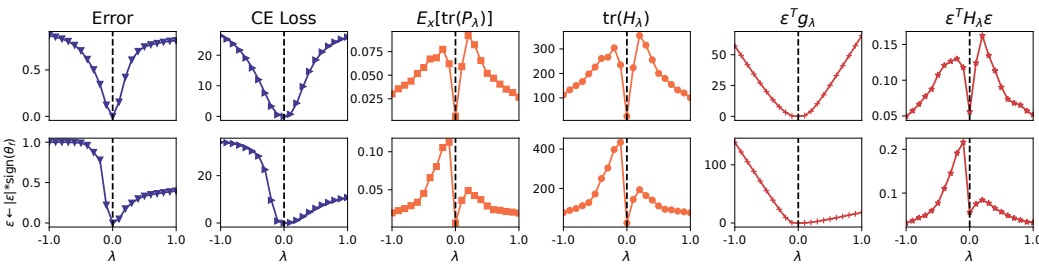

Figure 29: Several metrics calculated on a simple softmax classification demo.

```python
1  import numpy as np
2  from scipy.special import softmax
3  from matplotlib import pyplot as plt
4  from sklearn.datasets import load_digits
5  from sklearn.linear_model import LogisticRegression
6  from sklearn.metrics import log_loss
7
8  def grad_hessian(W, X, Y, Z, P):
9      N, D = X.shape
10     C = len(np.unique(Y))
11     Id = np.tile(np.diag(np.ones(C)), (N, 1, 1))
12     OY = np.diag(np.ones(C))[Y]
13     XX = (X[:, None, :] * X[:, :, None]).reshape(N, D * D)
14     PIP = (P[:, None, :] * (Id - P[:, :, None])).reshape(N, C * C)
15     g = -1.0 * np.dot((OY - P).T, X) / N
16     H = np.dot(PIP.T, XX) / N
17     H = H.reshape(C, C, D, D).transpose(0, 2, 1, 3).reshape(C * D,
           C * D)
18     return g, H
19
20 X, Y = load_digits(return_X_y=True)
21 model = LogisticRegression(max_iter=500, fit_intercept=False)
22 model.fit(X, Y)
23 W = model.coef_
24 noise_W = np.random.randn(*W.shape)
25 sign_noise_W = np.abs(noise_W) * (2.0 * (W > 0.0) - 1)
26
27 for noise in [noise_W, sign_noise_W]:
28     data = []
29     for lamb in np.linspace(-1.0, 1.0, 21):
30         Wn = W + lamb * noise
31         Z = np.dot(X, Wn.T)
32         P = softmax(Z, axis=1)
33         g, H = grad_hessian(W, X, Y, Z, P)
34         trp = np.sum(P * (1 - P), axis=1).mean()
35         trh = np.sum(np.diag(H))
36         fd = np.sum(g * Wn)
37         sd = np.dot(Wn.reshape(-1), np.dot(H, Wn.reshape(-1)))
38         loss = log_loss(y_true=Y, y_pred=P)
39         err = np.mean(np.argmax(P, axis=1) != Y)
40         data.append([trp, trh, fd, sd, loss, err])
41     data = np.array(data)
42     titles = ["GiniP", "Tr(H)", "n^Tg", "n^THn", "Loss", "Error"]
43     plt.figure(figsize=(15, 2))
44     for i in range(6):
45         plt.subplot(1, 6, i + 1)
46         plt.plot(data[:, i])
47         plt.title(titles[i])
48     plt.show()
```

