# OpenReview forum: "Exploring and Exploiting the Asymmetric Valley of Deep Neural Networks"
_NeurIPS.cc/2024/Conference — NeurIPS 2024 poster_

### Official Review · Reviewer_QEpC · 2024-07-06

**Soundness:** 3
**Presentation:** 3
**Contribution:** 3
**Rating:** 7
**Confidence:** 4

**Summary:**

In this work, the phenomenon of asymmetric valleys in deep neural networks (DNNs) minima, first observed and described in [1], is systematically examined. By studying different types of (random) directions around a given minimum, the authors discovered that the degree of sign consistency between the chosen direction and the minimum point is a key factor for asymmetry. Besides thorough empirical evaluation, theoretical reasoning is provided to explain this phenomenon. The findings are also practically applied in model fusion.

[1] Haowei He, Gao Huang, and Yang Yuan. Asymmetric valleys: Beyond sharp and flat local minima. In Advances in Neural Information Processing Systems 32, pages 2549–2560, 2019.

**Strengths:**

All in all, I like this work.

Generally better loss landscape understanding is important for modern deep learning, and this work takes another valuable step in this direction. To my knowledge, the question of minima asymmetry was not systematically investigated after its first appearance in [1], so I consider the contributions of this study novel and significant.

The paper is well structured. The narrative is easy to follow, the authors motivate their experiments (“Then, a fantastic idea motivates us to change the sign of the noise” – liked that) and consequently validate their conjectures.

The claims are clearly formulated and well supported empirically. Empirical evaluation is solid, involving multiple datasets, architectures, and hyperparameter options. The authors tackled many important questions concerning the study of asymmetric values and the sign consistency conjecture. Although theoretical support is mostly intuitive, with no rigorous proofs (the authors mention that in the Limitations section, which is commendable), it paves the way for future work that will address this limitation. Demonstrated practical implications of the insights of this work for model fusion are also important and require further investigation.

[1] Haowei He, Gao Huang, and Yang Yuan. Asymmetric valleys: Beyond sharp and flat local minima. In Advances in Neural Information Processing Systems 32, pages 2549–2560, 2019.

**Weaknesses:**

I see no concrete substantial weaknesses in this work. I list some of my questions and suggestions in the following section.

To me, the main shortcoming is weak theoretical support of the claims made in this work. However, the authors acknowledge this, and I hope that future work will address this disadvantage in some way.

Also, the experiment in Sec. 6.2 about practical implication in federated learning appears to me a little hastily conducted. Some more details would be appreciated (see specific questions below).

The text could be slightly polished to eliminate typos and small inaccuracies.

**Questions:**

1. The Investigation of 6 Special Noise Directions:
Empirically, noise directions that have the same sign with the positive parameters and applies zero to negative parameters ($\epsilon_5$ and $\epsilon_6$) do not show asymmetry. I wonder, why so? Do the authors have any intuition on that?
To me, that’s a little counterintuitive, since these directions specifically target the sensitive BN.weight parameters (according to the results of Sec. 4.2.1), don’t they? Also, does the same lack of asymmetry is exhibited by the negative-sign noise directions?
2. The Finding Holds for ImageNet and Various Parameter Groups:
    a) What type of noise is applied here?
    b) It is confusing that BN parameters seem like the most “robust to asymmetry” (according to Fig. 6) in light of other experimental results of this paper, underlining the specific role of BN parameters in asymmetry. Even the original work [1] claimed that “directions on BN parameters are more asymmetric”. Could the authors elaborate more on that?
3. Line 186: centered around 0.5?
4. Section 4.2.1:
Is the noise applied to all parameters or to BN parameters only? If to all, it is indeed remarkable how the initial distribution of BN weights, which take a negligible part in total model parameters, may influence the asymmetry of certain directions (getting back to Question 2b)!
5. Section 5: Did the authors train a plain linear classifier or a neural network in the sklearn.digits experiment?
6. Section 6.2:
Eq. (1): I might be wrong, but it seems that the regularizer is maximized when the signs of $\theta_t$ and $\theta_t^k$ are aligned, which is undesired, isn’t it?
The experiments here require stds, as the results presented in Table 1 could lack statistical significance.
Also, what are the hyperparameters of the baselines, how were they chosen? Were they validated just as $\gamma$ hyperparameter in FedSign?
7. I suppose that [2], where sign disagreement is also taken into account in model merging, could be a valuable related work.

[1] Haowei He, Gao Huang, and Yang Yuan. Asymmetric valleys: Beyond sharp and flat local minima. In Advances in Neural Information Processing Systems 32, pages 2549–2560, 2019.

[2] Yadav, Prateek, et al. Ties-merging: Resolving interference when merging models. Advances in Neural Information Processing Systems 37 (2023).

**Limitations:**

The authors discuss the limitations in Section 7.

---

> ### Author Rebuttal · Authors · 2024-08-06
>
> Thanks for reviewing our paper and finding our paper "**well structured and easy to follow**" and stating "**the claims are clearly formulated and well supported empirically**".
> Also, the authors thank you for **your recognition that the contributions are novel and significant when compared to the previous work [24]**.
>
> ########
>
> Q1: The Investigation of 6 Special Noise Directions.
>
> A1: A great question!
> Our paper's major observation is that **the sign consistency between DNN parameters and noise determines the symmetry**.
> Through your question, the authors guess that the level of asymmetry may **be also related to the property of perturbed parameters, e.g., the total number and distribution of the parameters**.
>
> To answer Q1, the special noise $\epsilon_5$ and $\epsilon_6$ does not show asymmetry because it only affects the positive parameters.
>
> If the parameters are all positive, they may be relatively stable when adding or subtracting some specific values.
> Otherwise, if the distribution of parameters is more disordered (e.g., containing various scopes of positive and negative values), the parameters may be more sensitive to sign-consistent perturbation.
> We investigate the negative-sign noise directions and find similar results.
>
> This guess may also provide explanations for BN layers, i.e., Q2.
>
> ########
>
> Q2: The Finding Holds for ImageNet and Various Parameter Groups.
>
> A2: First, we use Gaussian noise.
> Second, compared with other groups of parameters, **BN parameters (especially the BN.weight) have a simpler parameter distribution**.
> According to the first question's guess (e.g., A1), BN parameters tend to be less sensitive to sign-consistent noise.
>
> This observation does not contradict the findings in [24].
>
> First, they find that adding random noise directions on BN parameters is more asymmetric, which is due to the possible larger sign consistency.
>
> Second, adding totally sign-consistent noise to BN parameters indeed leads to asymmetry valleys (e.g., Fig.6 and Fig.7), but the asymmetry tendency is less obvious.
>
> Third, applying totally sign-consistent noise to other parameters (e.g., a complex parameter distribution) leads to obvious asymmetry valleys.
>
> As an initial guess, **the significance level of asymmetry has the following rank: A > B > C > D**.
>
> A denotes a complex parameter distribution perturbed by sign-consistent noise. (100% sign consistency)
>
> B denotes a simpler parameter distribution perturbed by sign-consistent noise. (100% sign consistency, but the parameter distribution is simpler)
>
> C denotes a simpler parameter distribution perturbed by random noise. (a larger probability of > 50% sign consistency)
>
> D denotes a complex parameter distribution perturbed by random noise. (about 50% sign consistency)
>
> The previous work [24] found **C > D** (i.e., directions on BN parameters are more asymmetric), while our paper found **A > D** and **B > C** (i.e., sign-consistent noise leads to asymmetry).
>
> Your question helps us guess that **A > B**, which derives the entire rank.
>
> In summary, your great question brings us a novel guess about the influence of parameter distribution on valley symmetry. This guess should be checked further. Thanks for your insights.
>
> ########
>
>
> Q3: Line 186.
>
> A3: We are sorry that this is a mistake. The distribution is centered around 0.5, while the ones under the initialization of U(0, 1) are centered around 0.2.
>
> ########
>
> Q4: Section 4.2.1.
>
> A4: We only apply noise sampled from \{0, 1\} to BN parameters, which conforms to the experimental setting in [24].
>
> We also have some experimental studies that apply noise to the whole parameters when considering the influence of BN, and please see the details in Appendix A.4.
> Specifically, we compare the valley symmetry of DNNs with or without BN layers.
>
> ########
>
> Q5: Section 5: Did the authors train a plain linear classifier or a neural network in the sklearn.digits experiment?
>
> A5: As we stated in Section 5 (line 260), the demo code is provided in Appendix D.3. The demo code is listed in Python code 2 (Page 23).
>
> The code trains a plain linear classifier as a simple illustration.
>
> ########
>
> Q6: Section 6.2
>
> A6: Thanks for your correction! The design of this loss follows the idea of Negative Log Likelihood, where we miss a minus. We will correct this formula in the future version.
>
> We rerun the experimental studies five times and list the std. of accuracies for a portion part in Table. 1. The experimental results are as follows. The accuracy doesn't fluctuate very much.
>
> |Dir.$\alpha$|FedAvg|FedSign|
> |-|-|-|
> |CIFAR-10 10.0|81.53 (0.17)|82.59 (0.09)|
> |CIFAR-10 1.0|80.54 (0.11)|80.76 (0.14)|
> |CIFAR-10 0.5|77.69 (0.21)|78.41 (0.35)|
>
> All results will be added in the future version.
>
> FedAvg does not have additional hyper-parameters. FedProx, MOON, and FedDyn have a regularization coefficient like $\gamma$, and we search for them in the scope of \{0.1, 0.01, 0.001\}. The best results are reported. The hyper-parameters of FedPAN are set according to the original paper.
>
> ########
>
> Q7: A valuable related work.
>
> A7: Thanks for your recommendation! This is indeed related to our work.

---

> > ### Comment · Reviewer_QEpC · 2024-08-11
> > **Reviewer's response**
> >
> > I thank the authors very much for the detailed and interesting commenting on my questions!
> >
> > I would like to remain my score unchanged and vote for acceptance.

---

### Official Review · Reviewer_Jp1U · 2024-07-11

**Soundness:** 3
**Presentation:** 3
**Contribution:** 2
**Rating:** 4
**Confidence:** 5

**Summary:**

This paper explores the factors affecting the symmetry of DNN valleys, encompassing (1) the dataset, network architecture, initialization, and hyperparameters that influence the convergence point; and (2) the magnitude and direction of the noise for 1D visualization. The major contribution is the observation that the degree of sign consistency between the perturbation and the convergence point is a critical indicator of valley symmetry. Theoretical insights from the aspects of ReLU activation and softmax function are provided to explain the asymmetry of DNN valleys. Imposing sign alignment in federated learning is proposed for model parameter alignment.

**Strengths:**

The observation that the degree of sign consistency between the noise and the convergence point is a critical indicator of valley symmetry is reasonable, and supported by the theoretical explaination based on ReLU activation and softmax function.

**Weaknesses:**

1. The asymmetry of DNN valleys is not unexpected, and has been studied in [24].
2. The observation that sign consistency between the noise and the convergence point affects the valley asymmetry, taking the ReLU activation and softmax function into account, is intuitive to a large degree and hence deos not emerge as a significant contribution.
3. Lack of theoretical analysis, such as the bound of degree of asymmetry caused by sign consistency.
4. Although the application of sign consistency in federated learning is provided, the readers may be more interested in the implications of valley asymmetry to local search based optimization methods, such as SGD, SAM etc.

**Questions:**

1. Fig.1 is not clear enough.
2. There are two "then"s in line 177.

**Limitations:**

Lack of theoretical analysis and the discussion on the implications of valley asymmetry to local search based optimization methods.

---

> ### Author Rebuttal · Authors · 2024-08-06
>
> Thanks for reviewing our paper and **finding our observation reasonable**.
>
> ########
>
> Q1: The asymmetry is not unexpected and has been studied in [24].
>
> A1: Surely, our work is majorly motivated by the proposal of asymmetry valley in [24], and we have declared this relation in lines 28-29 and 67-70.
> We have also declared that "**the asymmetric valley is initially proposed by [24], while it does not propose the inherent principles behind the phenomenon**" in Appendix A.4.
> Appendix A.4 also **in-detail presents our motivation for changing the sign of noise to explain the asymmetry valley**, which has not been studied in previous works.
> Compared with [24], we have the following additional contributions:
>
> (1) we provide **both empirical explanations and theoretical insights for the asymmetry valley**;
>
> (2) we provide **a novel concept named sign consistency ratio**;
>
> (3) we provide **detailed analysis for the DNNs with BN layers and show the initialization matters**;
>
> (4) we also provide **specific applications to model fusion**, including model soups and federated learning.
>
> Additionally, the reviewer QEpC has recognized that "**to my knowledge, the question of minima asymmetry was not systematically investigated after its first appearance in [24], so I consider the contributions of this study novel and significant**".
>
> ########
>
> Q2: The sign consistency is intuitive and does not emerge as a significant contribution.
>
> A2: We don't agree with the reviewer's view.
> When reviewers think that this explanation is intuitive and reasonable, they recognize the comprehensibility and acceptability of our work.
>
> However, they do not know the process of discovering this explanation required a lot of experimental studies, such as the process in Appendix A.4 and the extensive visualization results in the paper.
> These experiments include validation results on multiple data sets, various DNNs, different initialization methods, and even different DNN blocks on ImageNet.
>
> The authors advocate that **summarizing a reasonable explanation that is easy to understand and accepted by the community through abundant experimental results in various scenarios is a significant contribution to the research area**.
>
> ########
>
> Q3: Lack of theoretical analysis.
>
> A3: The lack of strict theoretical proof is indeed our weakness, but we provide some theoretical insights to support our findings.
>
> The reviewer MUA2 thinks that "**the theoretical insights are convincing**".
>
> The reviewer QEpC also declares that "**although theoretical support is mostly intuitive, with no rigorous proofs (the authors mention that in the Limitations section, which is commendable), it paves the way for future work that will address this limitation**".
>
> We will try our best to provide formal theoretical results in future work.
>
> ########
>
> Q4: The application to local-search based optimization methods, such as SGD and SAM.
>
> A4: Surely, the traditional works focus on searching for a wider minima that has better generalization performance, which has been stated in lines 76-79.
> However, these previous works majorly assume the width of minima is well correlated with the generalization performance, **which is not inclusive until now**.
>
> Hence, we provide **a novel application area** (i.e., model fusion) and apply our findings to federated learning.
> Additionally, our application to federated learning is well-motivated, which **is also recognized by the reviewer MUA2 and the reviewer QEpC**.

---

> > ### Comment · Reviewer_Jp1U · 2024-08-13
> >
> > Thank the authors for their replies. I still think this paper can be improved in such aspects as theoretical analysis and insights on its possible applications to local search based optimization. The authors stated theoretical insight is one of the main differences w.r.t literature [24], so I think a formal analysis behind the theoretical intuitions would be important, and the bound of degree of asymmetry, i.e., how large can the asymmetry be and the factors affecting it, is not addressed in the replies. Sharpness-aware minimization etc. utilize the sharpness to improve performance, so I would like to see a discussion on how the asymmetry could possibly be utilized to facilitate optimization or how it affects the optimization. This is also not addressed. Considering these, I  would like to maintain my score.

---

> > > ### Author Response · Authors · 2024-08-13
> > >
> > > Thanks for your further responses! Your suggestions are indeed valuable for our future work.
> > >
> > > Factually, our article **has already had some preliminary discussions** about your suggestions and concerns.
> > >
> > > ########
> > >
> > > Suggestion 1: About the theoretical analysis and the bound of the degree of asymmetry.
> > >
> > > Answer:
> > >
> > > (a) We have provided theoretical insights from the Hessian matrix of softmax classification in lines 251-260 and Appendix D.3.
> > >
> > > (b) About the degree of asymmetry, we utilized the average error on the left and right sides as a specific metric. This is shown in Fig. 4.
> > >
> > > Providing a formal theoretical analysis is the future work **as we stated in the Limitations section**.
> > >
> > > ########
> > >
> > > Suggestion 2: About the application to local search based optimization.
> > >
> > > Answer:
> > >
> > > (a) The related optimization works (e.g., Sharpness Aware Minimization [16] and Adaptive-SAM [40]) are all **based on the assumption that "flat minima leads to good generalization"**. Our explanation about asymmetry further makes this assumption get into dispute because different noise directions lead to valleys with different shapes, **making the calculation of flatness/sharpness uncertain and non-unique**. Hence, we do not focus on local search based optimization.
> > >
> > > (b) On the other hand, our application to federated learning **could also be viewed as a better optimization strategy**, which is to **enhance the ability of model fusion during the local training procedure** in federated learning.
> > >
> > > (c) If we really want to apply the findings of this paper to local search based optimization, we can replace the Gaussian noise disturbance used in traditional flatness/sharpness measures with a sign-consistent Gaussian noise, and **design an appropriate flatness/sharpness measure**, and then evaluate whether this appropriate flatness/sharpness measure is related to generalization.
> > > If there is a certain correlation, it may inspire further optimization design. Moreover, this is not a simple verification process and requires a significant amount of future work.
> > >
> > > To conclude, considering that **a lot of work has been done on explaining, analyzing, and utilizing asymmetric valleys in this paper**, we authors believe that the workload of the suggestions made by the reviewer can be studied as separate work in the future.
> > >
> > > We hope these responses can further address your concerns.

---

### Official Review · Reviewer_MUA2 · 2024-07-12

**Soundness:** 3
**Presentation:** 3
**Contribution:** 2
**Rating:** 5
**Confidence:** 4

**Summary:**

This paper investigates the characteristics of the asymmetric valley in deep neural networks (DNNs) for classification. The authors perform a perturbation analysis around the local minima, considering the direction of the injected noise. The asymmetric valley demonstrates that DNNs exhibit smaller fluctuations with sign-aligned noise, while opposite noise dramatically increases the test error. Other interesting findings include: (1) Batch normalization initialization and hyperparameter choices affect the occurrence of asymmetry. (2)The ReLU activation and softmax function might explain the phenomenon. The authors also leverage the findings to explain the success of model fusion and propose a sign-regularized method for federated learning.

**Strengths:**

1.	The paper is well-structured and clearly presents its interesting findings. The authors start with empirical findings and further apply their discoveries to practical applications, including the model fusion trick and federated learning.
2.	The theoretical insights regarding the presence of ReLU and softmax are convincing.
3.	The proposed direction-aligned noise method is simple and effective, potentially inspiring further research in areas like transfer learning and federated learning.

**Weaknesses:**

1.	In Section 6.1, the authors claim that pre-training is important for the success of model fusion. However, the compared models have different architectures (VGG16BN vs. ResNet18) and are trained on different datasets (CIFAR10 vs. Flowers). This experimental setting weakens the credibility of the conclusion.
2.	Regarding the conclusion of Fig. 8, “The interpolation curve shows that the small batch training (i.e., λ = 0.0) lies in a sharper and nearly symmetric valley, while the large batch training (i.e., λ = 1.0) lies in a flatter but asymmetric valley,” I find this problematic. Fig. 9 already shows that the valley is asymmetric for BS=32, while Fig. 8 concludes that BS=32 is symmetric. For the possible explanation for the lower part of Fig. 8, my understanding is that BS=2048 has a more biased value distribution, while BS=32 is more zero-centered. Therefore, the same strength of interpolated noise affects BS=32 more. Besides, the degree of asymmetry is not well-defined in the paper, the conclusion based on this comparison might be problematic.
3.	The conclusion from Fig. 8 also conflicts with the statement in related work that “large batch size training may lead to sharp minima with poor generalization.”

**Questions:**

See weaknesses.

**Limitations:**

The discussion doesn't include transformer architecture.

---

> ### Author Rebuttal · Authors · 2024-08-06
>
> Thanks for reviewing our paper and finding our paper's strengths, e.g., "**the paper is well-structured and clear**", "**the theoretical insights are convincing**", and "**the proposed method is simple and effective**".
>
> ########
>
> Q1: The different architectures in Section 6.1.
>
> A1: As stated in lines 263-265, the pre-trained model could make the traditionally encountered barrier disappear when linearly interpolating models [55, 64].
> The previous works **have already shown** that pre-training is important for the success of model fusion [55, 64].
> Hence, our focus in Section 6.1 is not to support this conclusion further.
> Instead, we would like to **show the correlation between sign consistency and the model interpolation performance**.
> Using different DNN architectures could support the correlation convincingly.
>
> According to the reviewer's suggestion, we also **replace the pre-trained resnet18 with a random initialized one**, and we find the plots tend to be like the left part of Fig. 11 (i.e., becoming similar to the random initialized VGG16BN).
> We will **add these plots for better illustration in the later version**.
> These three groups of plots could better verify the correlation found in Section 6.1.
>
> ########
>
> Q2: The valley flatness of BS 32 and BS 2024 in Fig. 8 and Fig. 9.
>
> A2: To answer this question, we must declare two points of view.
>
> (1) **The valley width is closely related to the method of visualization**.
>
> This point of view has been stated multiple times in our paper, e.g., the lines 60-69, the lines 95-99, and the lines 198-200.
> Indeed, large batch training may lead to poor generalization compared to small batch training.
> However, whether the large batch training leads to sharp minima is **still inconclusive**.
>
> First, the minima width in the visualization depends on both the parameter scale of $\theta_f$ and $\epsilon$ (e.g., previous works [37, 42] found) and the direction of $\epsilon$ (e.g., our paper found).
> For example, the plot in Fig. 8 adds $\epsilon=\theta_{f_2} - \theta_{f_1}$ to the BS 32 solution $\theta_{f_1}$, while Fig. 9 adds the norm-scaled $\epsilon \sim G(0, 1)$ to the BS 32 solution $\theta_{f_1}$.
> Different noises have different parameter scales and directions, which makes the visualized shape inconsistent.
>
> Second, this phenomenon **could also be found** in the previous work [42], i.e., the Fig.2 vs. Fig.3 in [42].
> In other words, **the inconsistency of valley width in Fig.8 and Fig.9 is rational**.
>
> (2) **Our work does not focus on the valley width but on the valley symmetry**.
>
> We have also stated this point of view in lines 27-30, 69-70, 94-95, 106-107, 198-200, and 205-207, etc. The **inconsistency of valley width has been explained in [42]**. We find that the **valley symmetry between Fig.8 and Fig. 9 also differs a lot**, which is because of the sign consistency ratio as shown in the upper-right of Fig. 8.
>
> To conclude, perturbed by norm-scaled noise, BS 32 indeed lies on a flatter minima (Fig.3 in [42] and Fig.9 in our paper).
> However, perturbed by the same noise $\epsilon=\theta_{f_2} - \theta_{f_1}$, BS 32 lies on a sharper minima (Fig.2 in [42] and Fig.8 in our paper).
> This implies that the valley width is closely related to the visualization method, which further **makes the correlation between valley width and generalization performance not so clear**.
> Our work does not focus on this debate and aims to study the reasons behind valley asymmetry.
>
> ########
>
> Q3: The conclusion from Fig. 8 also conflicts with the statement in related work that “large batch size training may lead to sharp minima with poor generalization.”
>
> A3: Indeed, large batch training may lead to poor generalization compared to small batch training.
> However, whether the large batch training leads to sharp minima is **still inconclusive**, which is stated in lines 26-27.
> Additionally, the "conflict" **has already been explained** in [42] by proposing a proper filter-norm scaled visualization method.

---

### Author Rebuttal · Authors · 2024-08-06

Thanks for all the reviewers' efforts in reviewing our paper!

We are delighted that the three reviewers found our strengths.

The reviewer MUA2 advocates "**the paper is well-structured and clear**", "**the theoretical insights are convincing**", and "**the proposed method is simple and effective**".

The reviewer Jp1U **finds our observation reasonable**.

The reviewer QEpC thinks our paper is "**well structured and easy to follow**" and states "**the claims are clearly formulated and well supported empirically**". Additionally, the reviewer QEpC recognizes that **the contributions are novel and significant when compared to the previous work [24]**.

However, the reviewer Jp1U has some concerns about our paper, and we list the brief responses correspondingly.

########

1. The asymmetry of DNN valleys has been studied in [24].

We have listed our novel contributions in the paper and also provide them in the detailed responses to reviewer Jp1U.

The reviewer QEpC recognizes that **the contributions are novel and significant when compared to the previous work [24]**.

########

2. The observation of sign consistency seems intuitive.

We don't agree with the reviewer's view. The authors advocate that **summarizing a reasonable explanation that is easy to understand and accepted by the community through abundant experimental results in various scenarios is a significant contribution to the research area**.

########

3. Lack of theoretical analysis.

We have provided some theoretical insights to support our findings.

We have pointed out this weakness in the Limitations section.

The reviewer MUA2 thinks that "**the theoretical insights are convincing**".

The reviewer QEpC also declares that "**although theoretical support is mostly intuitive, with no rigorous proofs (the authors mention that in the Limitations section, which is commendable), it paves the way for future work that will address this limitation**".

########

4. The implications of valley asymmetry to local search based optimization methods.

**A novel application area** (i.e., model fusion) is a contribution of our paper **instead of a weakness**.

---

### Decision · Program_Chairs · 2024-09-25

**Decision:**

Accept (poster)

**Comment:**

The paper shows under what conditions a minima of the loss surface is asymmetric along a given direction, i.e. when it significantly deviates from a locally symmetric (usually parabolic) shape.

Two reviewers voted for recommendation (7 and 5 scores), while one voted for rejection (score of 4). The discussion phase has not led to reviewers reconciling their opinions.

The discover condition under which a minima is asymmetric seems rather natural and simple, which I see as a strength of the paper.

The applications to model fusion and federated learning are intriguing, as also highlighted by QEpC and underscore the relevance of the studied concepts. For example, whether or not the minima is asymmetric along the interpolation line has a meaningful impact on how easy or hard it is to fuse models (”fine-tuning the pre-trained model does not change the parameter signs a lot, which facilitates the following parameter interpolation”)

Jp1U who voted for rejection remarked that the paper lacks theoretical results (e.g. lack of bounds of the degree of asymmetry). I think that while the paper does emphasizes empirical results and intuitions, it is not in itself a reason to reject the paper.

All in all, I think the paper, while borderline, does clear the bar for acceptance and it is my pleasure to recommend it for this year's program.